

# 1 Influence of semi-volatile aerosols on physical and optical

# 2 properties of aerosols in the Kathmandu Valley

**3 Sujan Shrestha[1,2], Siva Praveen Puppala[1], Bhupesh Adhikary[1], Kundan Lal Shrestha[2],**

**4 Arnico K. Panday[1]**

[1]International Centre for Integrated Mountain Development (ICIMOD), Khumaltar, Lalitpur, Nepal
[2]Kathmandu University, Department of Environmental Science and Engineering, Dhulikhel, Kavre, Nepal
*Correspondence to:* Siva Praveen Puppala (SivaPraveen.Puppala@icimod.org) and Sujan Shrestha
(Sujan.Shrestha@icimod.org and sujanshrestha101@gmail.com)

**10 Abstract**

A field study was conducted in the urban atmosphere of the Kathmandu Valley to study influence of the semi-volatile
aerosol fraction on physical and optical properties of aerosols. The study was carried out during the pre-monsoon
season of 2015. Our experimental setup consisted of a single ambient air inlet from which the flow was split into two
sets of identical sampling instruments; the first set was connected directly with an ambient sample while the second
set received the air sample through a thermodenuder (TDD). Four sets of experiments were conducted for our study
to understand aerosol number, size distribution, absorption, and scattering properties using Condensation Particle
Counter (CPC), Scanning Mobility Particle Sizer (SMPS), Aethalometer (AE33) and Nephelometer respectively. The
influence of semi-volatile aerosols were calculated based on the difference of aerosol properties at room temperature,
50°C, 100°C, 150°C, 200°C, 250°C and 300°C through set TDD temperatures to ambient sample. Our results show
that with increasing TDD temperature, the evaporated fraction of semi-volatile aerosols also increased. At room
temperature the semi-volatile fraction of aerosol number was 12%, while at 300°C it was as high as 49% of ambient
aerosol. Aerosol size distribution analysis from SMPS shows that with an increase in temperature from 50°C to 300°C,
the peak mobility diameter of particles shifted from around 60nm to 40nm. However, no distinct change in the
effective diameter of the aerosol size distribution was observed with increase in set TDD temperature. The change in
size of aerosols due to loss of semi-volatile component had a stronger influence (~70%) at larger size bins when
compared to (~20%) at smaller bins of SMPS. At 300°C, the semi-volatile aerosols amplified BC absorption by
approximately 28% while scattering by the semi-volatile aerosols contributed up to 71% of total scattering. The
Scattering Angstrom Exponent (SAE) of the semi-volatile aerosol fraction was found to be more sensitive at lower
temperatures (<100°C) than at higher temperatures. However the Absorption Angstrom Exponent (AAE) of the semi-
volatile aerosol fraction did not show any significant temperature dependence.
Keywords: semi-volatile aerosols, aerosols, aerosol number concentration, aerosol size distribution, absorption,
scattering, black carbon, Angstrom Exponent, Kathmandu Valley.





## 1    Introduction

Aerosols are suspended particles in the air that are solid, liquid or a mixture of both states,  with sizes ranging from a few nanometers to several micrometers *(Warneck, 2000; Zellner, 1999)*. Studies have shown that aerosols have profound effects on human health *(Kampa & Castanas, 2008; Mauderly & Chow, 2008)*, climate *(Pöschl, 2005)* and visibility of the atmosphere *(Dzubay et al., 1982)*. Atmospheric aerosols are among the key factors that influence the earth's radiative energy balance. Aerosols affect climate directly by absorption and scattering of incoming solar radiation *(Haywood & Shine, 1997; Yu et al., 2006)* and indirectly by acting as a cloud condensation nuclei and affecting the optical properties and life cycles of cloud *(Ishizaka and Adhikari, 2003)*.

Aerosols are classified as primary or secondary depending upon their origin. Primary aerosols are particles that are emitted directly, from the combustion of fossil fuels and biomass, such as black carbon as well as windblown mineral dust and sea salts. Secondary aerosols are formed due to condensation, oxidation and chemical transformation *(Seinfeld & Pandis, 2006)*. Secondary aerosols tend to be semi-volatile in nature *(Hennigan et al., 2008)*. The aerosol components that do not condense under normal atmospheric conditions are considered as volatile aerosols and those remaining in condensed phase under certain atmospheric conditions are termed as semi volatile aerosols. Whereas, the non-volatile aerosols have negligible vapor pressure and remains in condensed phase under normal atmospheric conditions *(Fuzzi et al., 2006)*. Semi-volatile aerosols are believed to contribute the most to the toxicity of particles *(Stevanovic et al., 2015)*.  The volatility of an aerosol gives an indication about its emission sources, history and chemical composition *(Capes et al., 2008)*. The semi-volatile fraction of an aerosol largely depends on source of aerosol generation and the atmospheric conditions around the sampling site *(Robinson et al., 2007)*.

Past field and laboratory measurements show that the volatility of aerosol is largely influenced by reaction temperature and precursor gases. Previous studies show that a large fraction of aerosols is highly volatile under 150°C *(Ishizaka and Adhikari, 2003; Murugavel and Chate, 2011 and references therein)*. Measurements made by *Lee et al. (2010)* and by *Murugavel and Chate (2011)* indicated that the semi-volatile fraction in ambient aerosol was between 50 and 80% of the number concentration.  *Lin (2013)* reported that the potential radiative effect of secondary organic aerosol (SOA), which are a major fraction of semi-volatile aerosols, on climate was around one third of total aerosols. *Chung & Seinfield (2002)* reported that organic carbon (OC) has a global radiative forcing of -0.09 to -0.17 Wm$^{-2}$ where 50% of the radiative forcing was contributed by SOA.  The above studies show that semi-volatile aerosols plan a significant role at local to global scales.

Various epidemiological studies have reported the impact of semi-volatile aerosols on human health *(Dalton et al., 2001; Ronai et al., 1994)*. Some of these studies have found that in higher concentrations these semi-volatile aerosols can act as potential carcinogens. For example, the semi volatile  polycyclic-aromatic hydrocarbons (PAHs) are not just carcinogenic but also causes genetic susceptibility and oncogene activation *(Ronai et al., 1994)*. The exposure of semi-volatile components, such as dioxins, induces heart diseases leading towards mortality *(Dalton et al., 2001)*. These studies point towards the need of critical assessment of the semi-volatile aerosol fraction thereby leading to the better understanding of human health end points.




The Kathmandu Valley in Nepal is a polluted city in South Asia. Rapid urbanization, uncontrolled increase in the
number of vehicles, dusty roads and the topography of the valley itself are the main causes of the deteriorating air
quality. Emissions from heavy traffic movements, brick-kilns, open burning of solid waste, as well as the arrival of
emissions from households and industrial activities outside the valley are responsible for high human exposure to
particulate matter. Several studies were conducted in past on black carbon (BC), $PM_{2.5}$ and $PM_{10}$ *(Aryal et al., 2009;*
*Majumder et al., 2012; Putero et al., 2015; Sharma et al., 2012)* to characterize the Kathmandu valley pollution. For
example *Sharma et al. (2012)* reported that daily mean concentration of BC during winter in the Kathmandu valley
reached as high as 39.9 μg m$^{-3}$ with annual average concentration 8.6±4.4 μg m$^{-3}$. *Majumder et al. (2012)* reported
average $PM_{10}$ concentration at ten different traffic intersection of 1076±316 μg m$^{-3}$ whereas *Aryal et al. (2009)*
reported $PM_{2.5}$ concentration of 90±24 μg m$^{-3}$. However, none of these studies assessed the contribution of the semi-
volatile aerosols. To the best of our knowledge, this is the first study to report the contribution of the semi-volatile
aerosols on physical and optical properties of aerosols in the Kathmandu Valley.

**2     Sampling Location and Period**
The Kathmandu Valley located (Fig. 1a) is at an elevation of about 1300m above mean sea level and is surrounded
by hills as high as 2500m as shown in Fig. 1b. We conducted our experiments to characterize the semi-volatile fraction
of aerosols contributing to the particulates of the valley. Experiments were conducted at the rooftop of the Integrated
Centre for International Mountain Development (ICIMOD) headquarter in Lalitpur, Nepal (27.6464° N, 85.3235° E).
As shown in the Fig. 1b, sampling site is located at 7 km south west of the city center. Sampling site is mostly
surrounded by residential dwellings, hospital, educational institutes and brick kilns (nearest one being within 2 km
radius). There are no obstructing structures around the sampling site within a 50 meter radius. During the
measurement, instruments sampled ambient air from an inlet located 2m above the roof of the four story building (~15
m above the ground).

Measurements presented in this study were made during pre-monsoon season of the year 2015. As discussed
previously, higher levels of pollution in the Kathmandu Valley has been reported during the pre-monsoon seasons
*(Aryal et al., 2009)*. To use a single Thermodenuder (TDD) instrument, we set up a set of experiments with different
instrument pairs that were run sequentially. All measurements were carried out within the span of a few days during
which there was no rain and the meteorological conditions did not change much.

**3     Experimental Setup**
Our experimental setup consisted of an ambient air inlet that split to two sets of identical sampling instruments. The
first set of instruments were connected directly to an ambient sample inlet and the second set received air from the
sample inlet after passing through the Thermodenuder (TDD). A schematic diagram of the instrumental setup is
shown in Fig. 2. Four sets of experiments were conducted with this set up, using four different pairs of instruments.



In each experiment, the thermodenuder's temperature was changed over time to examine the fractional loss of the
semi-volatile aerosol fraction at increasing temperatures.  The summary of four sets of experimental setup and
respective sampling dates are summarized in Table 1.  It is important to note that the experiments had to be halted
due to massive earthquake in Nepal during April to May. In the preceding sections the term "wet" sample is used to
refer to ambient measurements while the term "dry" sample refers to measurements carried out with instruments
coupled with TDD. TDD temperatures were set at room temperature, $50^0$C, $100^0$C, $150^0$C, $200^0$C, $250^0$C and $300^0$C
in experiments 1, 3 and 4.  Similar experimental temperatures were used in previous *studies (Ishizaka and Adhikari,*
*2003; Jennings et al., 1994; Murugavel and Chate, 2011).* In experiment 2, we only examined the relationship
between particle size and volatility at room temperature, $50^0$C, $100^0$C, $200^0$C and $300^0$C.

**3.1     Instrument description**
The thermodenuder (TDD) used in this experiment is a Low-Flow (4 L min$^{-1}$) Thermodenuder Model 3065,
manufactured by Topas GmbH, Germany.  The TDD consists of two sections: one is desorption and the other
adsorption. It removes the semi-volatile fraction of ambient sample by thermal desorption using a heating element.
The semi-volatile fraction that is evaporated by thermal desorption is then adsorbed by the activated carbon which is
used as the working material in adsorption section. As per the manufacturer specification, the instrument is capable
of operating up to $400^0$C. However, we operated TDD only up to $300^0$C. The activated carbon was changed regularly
to ensure best working state of TDD. The structure and operation of the TDD is discussed more by *Madl et al. (2003)*.

The Condensation Particle Counters (CPC) used in this study were model 3775 manufactured by TSI Inc., USA. This
instrument can detect particle as small as 4 nm diameter and over a wide range of 0 to $10^7$ particles per cm$^3$. CPC was
operated with flow rate of 1.5 L min$^{-1}$. Butanol was used as the working fluid and instrument was used on auto drain
mode. Efficiency and operation of CPC model 3775 has been further discussed by *Hermann et al. (2007)*.

Scanning Mobility Particle Sizer (SMPS 3034, TSI Inc., USA) was used to measure the size distribution of aerosol
particles. The SMPS 3034 works by separating fine particles within the range of 10 to 487 nm based on their electrical
mobility. SMPS 3034 is a compact instrument with inbuilt Differential Mobility Size Analyzer (DMA) and
Condensation Particle Counter (CPC).We used neutralizer model number 308701 which is x-ray based in this study.
Operation of SMPS 3034 has been further discussed by *Hogrefe et al. (2006)*.

The Magee Scientific Aethalometer Model AE33 was used to study the black carbon (BC) concentration and aerosol
absorption in this study. During the whole measurement period the maximum attenuation limit was set at 100. The
instrument was operated at flow rate 2 L min$^{-1}$. Aethalometer model AE33 measures BC concentration at seven
different wavelengths *(Drinovec et al., 2015)* by using filter based light attenuation due to aerosol loading. The
manufacturer calibrated instrument measured light attenuation with soot particle loading. However, in real conditions
light attenuation reported by instrument represents all absorbing aerosols including BC, organic carbon and dust. We





followed *Drinovec et al. (2015)* methodology to convert BC concentration to absorption by using the relationship: BC
Absorption in $mm^{-1}$ = BC (in $ng\ m^{-3}$)* $10^{-3}$ * MAC (the mass absorption cross sectional values). MAC values for
specific wavelength are given by *Drinovec et al. (2015)*. The absorption at 880nm wavelength is usually represented
by BC absorption, as other particles like dust and organic carbon do not absorb or their contribution to absorption is
negligible at this wavelength *(Singh et al., 2014)*. However, at 370nm wavelength aerosol absorption is represented
by all three BC, organic carbon and dust *(Lim et al., 2014)*. Light absorbing organic aerosols are also known as brown
carbon *(Andreae & Gelencser, 2006; Bahadur et al., 2012)*.

The TSI integrating Nephelometer 3563 was used to measure aerosol scattering coefficient at three different
wavelengths: 450nm, 550nm and 700nm. We followed correction methodology given by *Anderson & Ogren (1998)*.
Nephelometer records both total scattering and backscattering coefficients, however, we report results from total
scattering. We operated the instruments for 24 hours at all previously reported TDD set temperatures. We followed
the methodology explained by *Anderson and Ogren (1998)* to minimize angular truncation error in the dataset using
the relationship: $\sigma_{corrected}$ =Correction factor (C) * $\sigma_{neph}$; where, C is correction factor, $\sigma_{neph}$ is scattering coefficient
reported by instrument and $\sigma_{corrected}$ is corrected scattering coefficient.

**3.2     Quality Control**
Before identical instruments were used in different set of experiments, collocated inter comparison studies were
conducted to estimate any biases within themselves. Identical instruments were operated with single inlet using Y
connector for 24 hours period with 1 min time resolution. Thereafter correlation between the identical instruments
were calculated. The slope being approximately equal to 1 and $r^2$ values approximately 0.99, no correction factors
were required.  (Fig. S1 included in supplemental material).

Leakage tests were conducted to check if any inlet pipelines or thermodenuder had leaks within the system. The main
inlet shown in Fig. 2, was connected to high efficiency particulate arrestance (HEPA) filter to verify any leakage in
sampling system. HEPA filters are known to be highly efficient to produce zero aerosol concentration with a minimum
99.7% of contaminants greater than 0.3 micron (MIL-STD-282 method 102.9.1). During the beginning of each set of
experiment, HEPA filter was connected to the main inlet to check for any leakages in the experimental setup. When
HEPA filter was connected, particle concentration readings in both the identical instruments dropped to zero as shown
in Fig. S2. This exercise also gave confidence that the instrument setup was proper and there were no leaks in the
system.

**4     Results and discussion**
Experimental results are summarized according to aerosol number concentration, size distribution and optical
properties in this section.

**4.1  Influence of volatility on aerosol number concentration**




As discussed in Section 3, the influence of semi-volatile aerosols on aerosol number was identified using the CPC and
CPC-TDD setup. During the first experiment, TDD was operated at room temperature by not providing power to TDD
thermal desorption section. This experimental setup will provide information about the semi-volatile aerosol loss due
to the dry activated carbon column in the TDD. Activated carbon in the adsorption section of TDD changes the
equilibrium state of the semi-volatile aerosols and leads to some evaporation even at room temperature *(Huffman et*
*al., 2009).* We observed 12% contribution to particle loss from the semi-volatile aerosols at room temperature. Similar
losses of 10% to 15% have also been reported in previous experiments *(Fierz, Vernooij, & Burtscher, 2007; Lee et*
*al., 2010; Stevanovic et al., 2015).* These losses are governed by various other factors that cannot be completely
avoided, such as particle loss due to sedimentation in micro sized particles as explained by *Burtscher et al. (2001)* and
particle loss due to thermophoretic and diffusional losses in submicron particles as mentioned by *Stevanovic et al.*
*(2015).*

The CPC and CPC-TDD setup as shown in Fig. 2 was operated for 24 hours for each set temperature.  The comparison
of wet and dry particle number concentration at TDD set temperatures of 50°C and 300°C are shown in Fig. 3a. Slope
of the scatter plot gives the fraction of dry particles at a given temperature. We can see in Fig. 3a that 16% and 49%
particle loss at 50°C and 300°C respectively. Strong correlation between wet and dry CPC indicates that the fraction
of the semi-volatile aerosols at different ambient particle concentration is similar. However, Fig. 3a also shows that
the correlation between wet and dry falls apart when the ambient particle number concentration is very high (>50000
#/cm$^3$). Although wet and dry particle number comparison is shown for 50°C and 300°C in Fig. 3a, similar results
were obtained for other TDD set temperatures. Result summary for other temperatures are given in Fig. 3b.

Figure 3b shows the temperature dependence of the semi-volatile fraction of aerosols. Using one minute data, we
calculated loss of particle percentage in dry sampling displayed in the box plots. The semi-volatile aerosol number
fraction was observed to be 12%, 16%, 18%, 23%, 28%, 46% and 49% at room temp., 50, 100, 150, 200, 250, 300°C
set TDD temperatures respectively.

*Murugavel & Chate (2011)* reported that at temperature below 150°C, 51%-71% particles evaporated out of ambient
aerosol during their experiment carried out in Pune. The authors further reported a 13%-26% loss between 150-300°C
and a 7%-13% loss of particle at temperatures greater than 300°C. These results show that the evaporated fraction of
the semi-volatile aerosols at different temperature ranges is comparatively less in Kathmandu than Pune. *Murugavel*
*& Chate (2011)* used SMPS to study particle loss in their study in Pune, India. Whereas, we used CPC for this study,
which measures particle number concentration from 4 nm to few microns. SMPS used by *Murugavel & Chate (2011)*
only measures particle number concentration between size range 10 nm to 487 nm. Kathmandu and Pune have
different source characteristics, topography and local meteorological conditions. Thus differences in results with
*Murugavel & Chate (2011)* are to be expected.





We compared the semi-volatile aerosol fraction from our experiment with the standard chemical compounds which
have evaporating temperature equivalent to TDD set temperatures. This comparison will provide a vital information
about aerosol chemical composition and volatility. The similar comparison technique has been adopted by others in
earlier studies as well *(Burtscher et al., 2001; Ishizaka & Adhikari, 2003; Murugavel & Chate, 2011)*.

For further analysis, we categorized the aerosol volatility into two categories: (I) highly volatile for those components
which vaporize at the temperature ≤150°C and (II) moderately volatile for those which vaporize between 150°-300°C.
We found 23% of aerosols to be highly volatile and 26% to be moderately volatile during the experimental period.
*Ishizaka & Adhikari (2003)* categorized ammonium chloride, ammonium sulphate, terpene, organic nitrogen, organic
matters, dioctyl phthalate, benzene, toluene, ethyl benzene, xylene, sulfuric acid, acetic acid and formic acid as highly
volatile components. The average aerosol number concentration during our experimental period was 16136 #/cm$^3$.
Out of this 2038 #/cm$^3$ are highly volatile in nature. These particles relatively may represent above mentioned highly
volatile aerosol components. Further *Ishizaka & Adhikari (2003)* categorized ammonium sulphate, ammonium
bisulphate, ammonium nitrate, diesel exhaust and secondary organic carbon as moderately volatile components. We
found 6319 #/cm$^3$ aerosol particles are moderately volatile in nature. In experiments carried out by *Ishizaka & Adhikari*
*(2003) and Shrestha et al. (2014)* the remaining particles which did not volatilize up to 300°C were soot carbon,
polymerized organic compounds, calcium carbonate, sea salt and mineral dust. Characterizing aerosol volatility as a
function of chemical composition in the Kathmandu Valley needs further investigation.

The diurnal variation of wet and dry aerosol number concentration and the semi-volatile aerosol fraction at 50°C and
300°C are shown in Fig. 4. We show results from only these two temperatures to represent minimum and maximum
TDD set temperatures. Results from other TDD set temperatures lie in between these two temperature results. Fig. 4
(a & b) show diurnal variation in wet aerosol number concentration which has one peak during morning hours around
9AM and the other peak during evening hours around 8PM. Previous studies by *Panday & Prinn (2009) and Putero*
*et al. (2015)* reported similar profile and also explained the physical processes and emission sources.

Figure 4 (c & d) shows the semi-volatile aerosol number fraction at 50°C and 300°C TDD set temperatures. The semi-
volatile aerosol fraction did not show any diurnal variation like wet aerosols at 50°C, which indicates that the semi-
volatile aerosol number fraction is uniform throughout the sampling period. However, as shown in Fig. 4d at TDD set
temperature 300°C the semi-volatile aerosol fraction changes with spikes in wet aerosol number concentration. These
spikes may be representing different air mass or fresh nearby sources. Thus, the semi-volatile aerosol number fraction
is significantly higher during peak events.

By comparing diurnal variation of highly and moderately volatile aerosol fractions, we noticed highly volatile aerosol
contribution was almost similar throughout the day while moderately volatile aerosol fraction changed significantly
during peak events (Fig. S3). As mentioned earlier, one of the moderately volatile aerosol source is diesel combustion
and hence the spikes in aerosol number concentration may be representing diesel combustion sources.





### 4.2    Influence of volatility on aerosol size distribution


The experimental setup using SMPSs was different from other instrumental setups. We operated identical SMPSs as
mentioned in the instrument setup section, but changing the TDD set temperature every hour. The purpose of this
experiment was just to understand particle size loss due to the semi-volatile aerosol fraction rather than diurnal
variability.

Results of SMPS measurements are summarized in Table 2. The semi-volatile aerosol number fraction was observed
slightly higher during SMPS experiment compared to CPC (see values in Table 1). The semi-volatile aerosol fraction
was observed to be 62% and 49% during SMPS and CPC experiments respectively at TDD set temperature 300°C.
Similar behavior was seen at other temperatures as well. One possible reason for the difference might be that CPC
measurements covered aerosol number concentrations from 4 nm to a few microns whereas the SMPS only covers
10nm to 487nm. The semi-volatile aerosol fraction may be higher in smaller diameter particles compared to larger
diameter particles. Figure 5 shows the number size distribution for TDD set at room temperature and 300°C. Even
though the dry aerosol number size distribution at room temperature was significantly lower to wet aerosol, their peak
mobility diameter did not shift much (from around 85nm to 80nm). Similar comparison at 300°C shows a different
result; here the peak diameter shifted from around 60nm to 40nm. *An et al. (2007)* reported that when individual
ammonium sulphate particle of different sizes were evaporated at different temperatures, the particle size decreased
significantly. This decrease in particle size of individual components of aerosol contributed in the shift in peak
diameter towards smaller diameter.

Individual size bin semi-volatile aerosols loss as a function of the particle diameter at all experimental temperatures
is shown in Fig. 6. Figure 6 shows a greater reduction of aerosol size for larger diameter aerosols compared to smaller
diameter aerosols. This may be because, the semi-volatile aerosol fraction at larger diameter may be in the form of a
coating or internally mixed state. By losing this fraction the aerosol size is expected to decrease. This is corroborated
by the observation that, as the size of the aerosols decrease, the peak diameter also shifted towards smaller diameter
at higher temperature. The shift in peak diameter was observed around 5-7nm between wet and dry sampling at TDD
set temperature ≤100°C, while this shift significantly change to  around 20-22nm at TDD set temperature 200°C to
300°C. Murugavel and Chate (2011), had results that are variable as the particle diameter increases. The difference
may be attributed to data collection, *Murugavel and Chate (2011)'s* reported data represents monthly and annual
average, whereas our present study represents an event sampling (one hour). Our results show that the mixing process
of ambient aerosols with highly and moderately volatile aerosols/precursors are different and needs further
investigation.

### 4.3    Influence of volatility on aerosol absorption

In real atmospheric conditions, BC exists in both elemental and in a mixed state with other compounds. Aethalometer
derived aerosol absorption represents both these states. Previous studies show that non-absorbing material coating
over elemental carbon (EC) enhances absorption due to the lensing effect *(Lack & Cappa, 2010; Schnaiter et al.,*



*2005; Shiraiwa et al., 2010; Zhang et al., 2008).* By removing this coating elemental carbon absorption will change.
Dust aerosol displays absorption features in the ultraviolet (UV) through the VIS wavelengths due to its mineralogical
composition, however dust aerosol is non-volatile in nature. Elemental carbon can be volatilized above 600°C
*(Shrestha et al., 2014),* but below 300°C it is stable.  In this study the semi-volatile aerosol absorption represented
only by either light absorbing organics or material (organic/inorganic) coated on EC.

The dust absorption is generally low in the spectral regime above 600 nm or tend to have constant background
absorption value for wavelengths larger than about 600 nm *(Gillespie & Lindberg, 1992; Lindberg, Douglass, &*
*Garvey, 1993; Sokolik & Toon, 1999)(Cao et al., 2005; Kumar, National, & Kanpur, 2008).* Hence, wet and dry
absorption measured from the aethalometer at 880nm wavelength is representative of either EC or mixed state of EC
absorption. Comparison of wet and dry aerosol absorption at 880nm for 50°C and 300°C is shown in Fig. 7a. The
semi-volatile aerosol absorption contribution to the total aerosol absorption was observed to be 20% and 28% at 50°C
and 300°C respectively at 880nm wavelength. Loss of absorption from 50°C to 300°C increased only 8%, which is
less compared to particle loss of around 33%. The particle loss was not proportional to the absorption loss at 880nm
wavelength mainly due to EC and its mixing state. Since EC and dust cannot be volatilized under these set
temperatures, the absorption contribution is coming only from mixing state at 880nm wavelength.

Highly and moderately volatile aerosol fraction contributed 21% and 7% respectively to aerosol absorption at 880nm
wavelength. As shown in Table 3, one fourth of aerosol absorption at 880nm wavelength is contributed by the semi-
volatile aerosols. Wet and dry aerosol absorption at 370nm is shown in Fig. 7b. The semi-volatile aerosol absorption
was slightly higher at 370nm compared to 880nm and the results are also tabulated in Table 3. The correlation for wet
and dry absorption at both wavelengths stays similar at a higher range unlike aerosol number concentration reported
by CPC in section 4.1. Results for other TDD set temperatures are summarized in Fig. 7c.

If we assume BC mixing state absorption affects is similar at 370nm and 880nm wavelengths, then the difference
between 370nm to 880nm wavelength absorption is contributed by brown carbon. This brown carbon absorption is
around 3% and 9% at 50°C and 300°C respectively. As shown in Table 3, highly volatile aerosol fraction does not
enhance much aerosol absorption at lower wavelengths (0-3%). This indicates the highly volatile aerosols are not true
representative of brown carbon aerosols. Further, as our results show, the moderately volatile aerosol fraction
absorption does enhance 4-9% at 370nm compared to 880nm. This indicates that brown carbon aerosols are
moderately volatile in nature. Results from Table 3 show that the brown carbon contribution is relatively less (0-9%)
compared to absorption enhancement (16-28%) due to EC mixing state with the semi-volatile fraction of aerosol
during our experimental period.

We report diurnal variation of absorption using the aethalometer and aethalometer coupled with TDD setup at 50°C
and 300°C for 520nm wavelength in Fig. 8 (a & b). As expected, both figures show increase in BC absorption during
the early morning hours, lowering during afternoon and finally building up again in the evening.  This is similar to





results from *Backman et al. (2012)* which explained similar BC diurnal variation. Figure 8c shows that at 50°C both
wet and dry aerosol absorption shows similar magnitude as well as diurnal variation.  The semi-volatile aerosol
absorption is around 20% at 50°C TDD set temperature. However, as shown in Fig. 8d, although the diurnal variability
is similar at 300°C, the semi-volatile aerosol absorption increased to around 30%. Unlike CPC, the semi-volatile
aerosol absorption is more variable throughout the day at TDD set temperature 50°C. It is also noteworthy that the
semi-volatile aerosol absorption show similar variability at both TDD set temperatures.

Further, the data obtained from aethalometer were used to find the wavelength dependency of absorption which is
usually expressed as an Absorption Angstrom Exponent (AAE). AAE is simply the negative slope of log of absorption
by log of two different wavelengths. In past studies, AAE has been used to infer about the dominant composition of
absorbing aerosol in the atmosphere *Bergstrom et al. (2007)*. Several studies reported AAE value close to 1 for fossil
fuel sources and around 2 for biomass sources *(Kirchstetter, Novakov, & Hobbs, 2004)*. Our results, as summarized
in Table 3 show wet aerosol AAE ranges from 0.97 to 1.30 with a median value around 1 implying sampled aerosols
are dominated by fossil fuel sources. The AAE results for 300°C shown in Fig. 9 was around 1.5 indicating the
influence of biomass burning source(s).

Dry absorption was deducted from wet absorption values to compute the semi-volatile aerosol absorption. Then semi-
volatile aerosol absorption values were used to compute AAE for the semi-volatile fraction. We computed AAE over
the range of wavelengths 370nm-970nm for wet, dry and semi-volatile aerosols individually and results for the entire
range of wavelengths and different TDD set temperatures are shown in Fig. 9. As shown in Table 3, the semi-volatile
aerosol AAE was observed in between 1.10 to 1.43. The maximum semi-volatile aerosol AAE value was observed
when the sample was influenced by biomass sources. Highly volatile aerosols AAE was observed around 1.1 whereas
moderately volatile aerosol AAE was observed in between 1.1-1.4. As discussed earlier the highly volatile aerosol
absorption was mainly from the BC mixing state. This mixing state AAE is around 1.1. *Lack and Langridge (2013)*
reported brown carbon AAE tend to be around 2-10. Our observed semi-volatile aerosol AAE was below this range
further corroborating the results that absorption is mainly influenced by mixing state as compared to brown carbon.



### 4.4    Influence of volatility on aerosol scattering

In this section, we discuss the influence of volatility on the scattering properties of the aerosols. As shown in Fig. 10a, the semi-volatile aerosol scattering contribution at 700nm wavelength was observed to be 8% and 66% of wet aerosol scattering at 50°C and 300°C TDD set temperatures. Whereas the semi-volatile aerosol scattering contribution increased to 17% and 71% at 450nm wavelength as shown in Fig. 10b. This wavelength dependency of scattering loss is evident for all set temperatures and results are plotted in Fig. 11a. Even though the CPC and scattering experiments were conducted on different days, by assuming the urban air mass characteristics remains similar, we infer that particle loss are not proportional to scattering loss as shown in Fig. 3a and Fig. 10 (a & b). However the scattering loss at 700nm wavelength was somewhat similar (66% to 62% versus 66% to 49%) to the particle loss in SMPS experiment as shown in Table 2 and Table 4. Thus the smaller particle semi-volatile aerosol fraction has greater influence in total aerosol scattering. The constraint of the experiment is that the TDD flow rate, which is restricted to 3 Lpm. Because of this we could not connect multiple instruments to maintain the TDD flow rate. More tests are required to statistically validate inferred results. Results are summarized for all three wavelengths (450nm, 550nm and 700nm) at different TDD set temperatures in Fig. 11a. Diurnal variability of wet and dry aerosol scattering (figure not shown) was observed similar to CPC experiment.

We computed Scattering Angstrom Exponent (SAE) similar to the methodology explained in sect. 4.3. The semi-volatile aerosol fraction SAE results plotted in Fig. 11b shows different results compared to AAE. The semi-volatile aerosol fraction AAE was almost similar at all TDD set temperatures while SAE shows higher values for room temperature and 50°C compared to other TDD set temperatures. The semi-volatile aerosol fraction SAE values were observed to be greater than 4 at room temperature and 50°C. This indicates scattering contribution of the semi-volatile fraction almost 8 times higher at 450nm wavelength compared to 700nm wavelength at room temperature and 50°C. For other TDD set temperatures, the semi-volatile aerosol fraction contribution to scattering is around 3 times at 450nm wavelength compared to 700nm wavelength, which is slightly higher than that of wet aerosol SAE (Table 4).

### 4.5    Influence of volatility on aerosol single scattering albedo

Single Scattering Albedo (SSA) is the ratio of scattering coefficient to extinction coefficient, which provides an indication of how absorbing or scattering the sampled aerosol is. The SSA value greater than 0.95 represents aerosol with a net effect of cooling whereas less than 0.85 will have a warming effect. The SSA values in between 0.85 and 0.95 may represent warming or cooling effect depending upon surface albedo and cloud cover *(Ramanathan, Crutzen, Kiehl, & Rosenfeld, 2001)*. We computed SSA values for different semi-volatile aerosol fraction at different TDD set temperatures by assuming wet aerosol SSA as 0.9 and 0.95. By assuming wet aerosol SSA as 0.9 we can derive scattering and absorption coefficient values as 100X and 11X (X can be any arbitrary value). We know from sections 4.3 and 4.4 the amount of scattering and absorption contribution from the semi-volatile aerosol fraction. By applying these fractions to 100X and 11X values we can retrieve scattering and absorption coefficients of the semi-volatile aerosol fraction and calculate SSA. By adopting this method, we calculated the semi-volatile aerosol fraction SSA values and are given in Table 5.




The results show semi-volatile aerosol fraction SSA was observed lower at room temperature and 50°C compared to
other TDD set temperature at all wavelengths. In addition, the semi-volatile aerosol fraction is more absorbing in
nature at 700nm compared to 450nm wavelength. This may be because scattering and absorption loss are not similar
at different wavelengths. Our results show at TDD set temperature 50°C, the semi-volatile aerosol fraction SSA was
observed to be minimum at all wavelengths. The semi-volatile aerosol fraction scattering loss was observed relatively
high compared to its absorption TDD set temperature at 50°C. This led to lower SSA values at this temperature. If
this process is applicable in atmospheric condition, noon time temperature rise may influence the net aerosol optical
properties and makes the atmosphere more absorbing in nature.

**5    Conclusion**
This is the first of its kind study to quantify the semi-volatile aerosol influence on aerosol physical and optical
properties over the Kathmandu Valley. Experimental results show that the semi-volatile aerosol number fraction
ranged from 12 to 49% at TDD set temperatures from room to 300°C respectively. During our experiment, we
observed that the highly volatile aerosols do not exhibit diurnal variability while the moderately volatile aerosols
contribution increases during peak concentration events. In addition, SMPS experiment results showed that the
reduction of the aerosol size was high for larger diameter aerosols compared to smaller diameter aerosols due to
removal of the semi-volatile aerosol fraction. Through our experimental results we noticed that the semi-volatile
aerosols mixing state contributed around 20% to total aerosol absorption. Aerosol absorption by the semi-volatile
aerosols were observed to be in between 16 to 28% at 880nm wavelength whereas calculated brown carbon
contribution to the aerosol absorption ranged from 0 to 9%. The scattering contribution was observed to be in the
range 18 to 71% and 8 to 66% at 450nm and 700nm respectively. Our results showed that the semi-volatile aerosol
contribution to aerosol scattering was significantly higher compared to aerosol absorption and number. Since the semi-
volatile aerosol scattering contribution was found to be two times higher than its absorption, implying removal of the
semi-volatile aerosols will lead to the more absorbing atmosphere.
Our study shows that the semi-volatile aerosols play important role in characterizing aerosol physical and optical
properties over the Kathmandu Valley. The results are discussed based on limited aerosol sampling and has scope for
future studies for better understanding seasonality, source and meteorological influence.
**Acknowledgements and Disclaimer**
This project was partially funded by core funds of ICIMOD contributed by the government of Afghanistan, Australia,
Austria, Bangladesh, Bhutan, China, India, Myanmar, Nepal, Norway, Pakistan, Switzerland, and the United
Kingdom. We thank Pradeep Dangol for his technical support during initial instrument setup. The first author is also
grateful to the Department of Environmental Science and Engineering, Kathmandu University for encouragement in
carrying out this study. The views and interpretations in this publication are those of the authors and are not necessarily
attributable to ICIMOD.



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



**Table 1.** Summary of four sets of experiments carried out with their respective sampling dates

| S.N. | Experimental setup | Experiment date |
|---|---|---|
| 1 | Semi-volatile aerosol contribution to particle number concentration using CPC and thermodenuder setup | March-April, 2015 |
| 2 | Semi-volatile aerosol contribution to aerosol size distribution using SMPS and thermodenuder setup | June, 2015 |
| 3 | Semi-volatile aerosol contribution to total aerosol absorption using aethalometer and thermodenuder setup | April, 2015 |
| 4 | Semi-volatile aerosol contribution to total aerosol scattering using nephelometer and thermodenuder setup | July, 2015 |




**Table 2.** Summary of semi-volatile aerosol fraction's physical properties at various temperatures.

| TDD set temp. in °C | Semi-volatile fraction of aerosol measured by CPC (%) | Semi-volatile fraction of aerosol measured by SMPS (%) |
|---|---|---|
| Room temp. | 12 | 20 |
| 50 | 16 | 26 |
| 100 | 18 | 32 |
| 150 | 23 | - |
| 200 | 28 | 52 |
| 250 | 46 | - |
| 300 | 49 | 62 |




**Table 3.** Summary of influence of volatility on absorption at various temperatures.

| TDD set temp. in ˚C | Loss of absorption at 370nm (%) | Loss of absorption at 880 nm (%) | Absorption by brown carbon | Average absorption angstrom coefficient of wet aerosol (Avg±SD) | Average absorption angstrom coefficient of dry aerosol (Avg±SD) | Average absorption angstrom coefficient of semi-volatile aerosol (Avg±SD) |
|---|---|---|---|---|---|---|
| Room temp. | 16 | 16 | 0 | 1.02±0.24 | 1.01±0.24 | 1.12±0.47 |
| 50 | 23 | 20 | 3 | 1.08±0.17 | 1.08±0.18 | 1.10±0.43 |
| 100 | 19 | 18 | 1 | 1.01±0.23 | 0.98±0.23 | 1.12±0.38 |
| 150 | 25 | 21 | 4 | 0.97±0.27 | 0.92±0.30 | 1.19±0.41 |
| 200 | 31 | 27 | 4 | 0.97±0.19 | 0.92±0.18 | 1.13±0.43 |
| 250 | 35 | 28 | 7 | 1.03±0.20 | 0.99±0.22 | 1.12±0.30 |
| 300 | 37 | 28 | 9 | 1.30±0.30 | 1.24±0.30 | 1.43±0.33 |







**Table 4.** Summary of influence of volatility on scattering at various temperatures.

| TDD set temp. in °C | Loss of scattering at 450nm (%) | Loss of scattering at 550nm (%) | Loss of scattering at 700nm (%) | Average scattering Angstrom coefficient of wet aerosol (Avg±SD) | Average scattering Angstrom coefficient of dry aerosol (Avg±SD) | Average scattering Angstrom coefficient of semi-volatile aerosol (Avg±SD) |
|---|---|---|---|---|---|---|
| Room temp. | 18 | 15 | 8 | 1.94±0.45 | 1.68±0.45 | 4.35±2.46 |
| 50 | 17 | 13 | 8 | 1.76±0.38 | 1.47±0.34 | 5.52±2.01 |
| 100 | 29 | 27 | 20 | 1.92±0.42 | 1.69±0.39 | 2.85±0.32 |
| 150 | 39 | 38 | 32 | 1.96±0.44 | 1.70±0.44 | 2.69±0.71 |
| 200 | 48 | 46 | 40 | 1.93±0.42 | 1.59±0.40 | 2.65±0.64 |
| 250 | 62 | 59 | 52 | 1.94±0.44 | 1.45±0.41 | 2.61±0.46 |
| 300 | 71 | 70 | 66 | 1.99±0.46 | 1.49±0.41 | 2.47±0.80 |




**Table 5**. Summary of semi-volatile aerosol fraction Single Scattering Albedo (SSA) assuming wet aerosol SSA as 0.9 and 0.95 at different wavelengths and
TDD set temperatures.

| TDD set temp. in ˚C | Wavelength = 450nm | | Wavelength = 550nm | | Wavelength = 700nm | |
|---|---|---|---|---|---|---|
| | Semi-volatile aerosol fraction Single Scattering Albedo (SSA) at wet aerosol fraction SSA 0.9 | Semi-volatile aerosol fraction Single Scattering Albedo (SSA) at wet aerosol fraction SSA 0.95 | Semi-volatile aerosol fraction Single Scattering Albedo (SSA) at wet aerosol fraction SSA 0.9 | Semi-volatile aerosol fraction Single Scattering Albedo (SSA) at wet aerosol fraction SSA 0.95 | Semi-volatile aerosol fraction Single Scattering Albedo (SSA) at wet aerosol fraction SSA 0.9 | Semi-volatile aerosol fraction Single Scattering Albedo (SSA) at wet aerosol fraction SSA 0.95 |
| Room temp. | 0.91 | 0.95 | 0.90 | 0.95 | 0.85 | 0.92 |
| 50 | 0.86 | 0.93 | 0.83 | 0.91 | 0.78 | 0.88 |
| 100 | 0.93 | 0.97 | 0.93 | 0.97 | 0.92 | 0.96 |
| 150 | 0.94 | 0.97 | 0.94 | 0.97 | 0.94 | 0.97 |
| 200 | 0.94 | 0.97 | 0.94 | 0.97 | 0.94 | 0.97 |
| 250 | 0.95 | 0.96 | 0.95 | 0.97 | 0.95 | 0.97 |
| 300 | 0.95 | 0.98 | 0.95 | 0.98 | 0.96 | 0.98 |









**Figures**




**Fig. 1. (a)** Satellite image of South Asia showing the location of Kathmandu Valley (indicated by red square
symbol). **(b)** Elevation contour map displaying the Kathmandu valley and the ICIMOD sampling site indicated by
symbol "*". Color bar indicates elevation above mean sea level in meters. The red square in the top figure (a) has
same coordinates as the bottom figure (b).



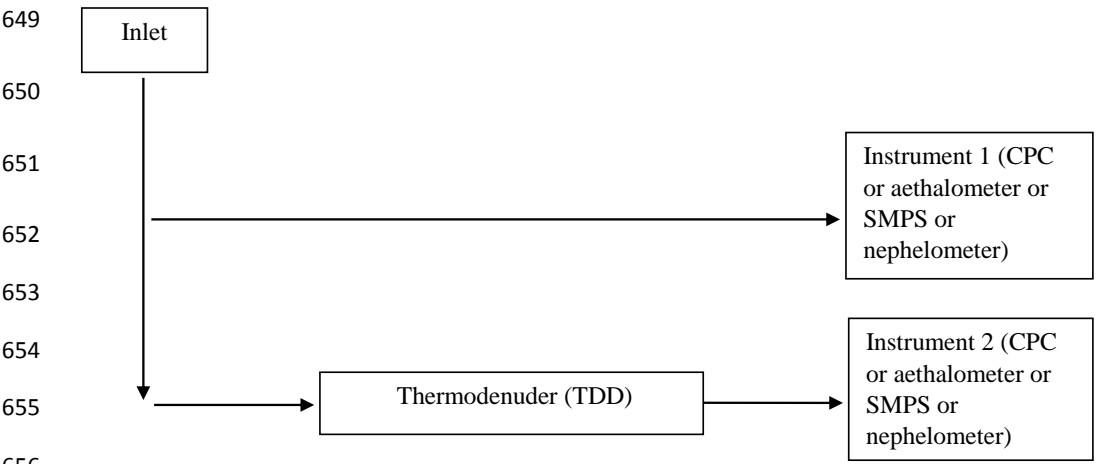









**Fig. 2.** Schematic of instrumental setup. CPC, aethalometer, SMPS and nephelometer were operated at flow rates
1.5 lpm, 2 lpm, 5 lpm and 5 lpm respectively. Identical instruments were maintained with same flow rates. At a
time, the experiment was carried out with only one set of instruments, i.e. either CPC or aethalometer or so on.














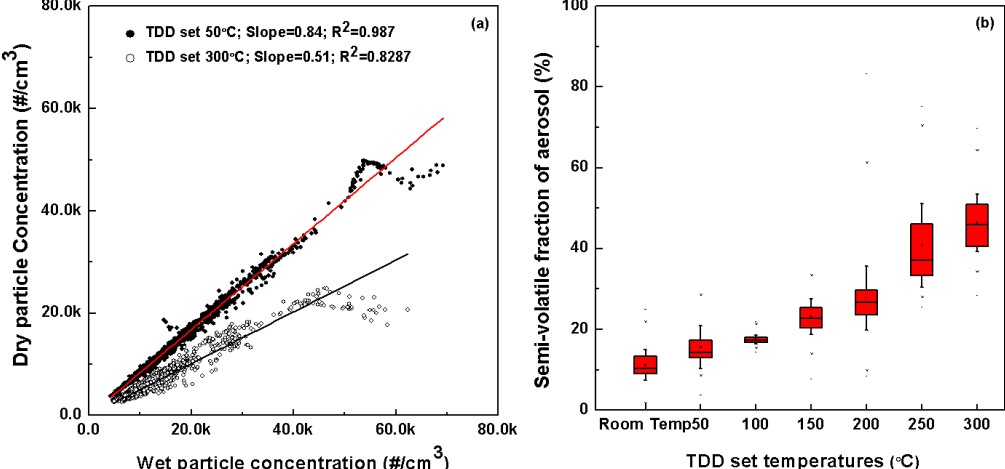


**Fig. 3. (a)** Comparison of wet versus dry particle concentration measured from CPC. Solid red and black lines

indicate the slope of linear regression analysis for TDD set temperatures for 50°C and 300°C respectively, **(b)**
Boxplot of measured semi-volatile fraction of aerosols at different TDD set temperatures.















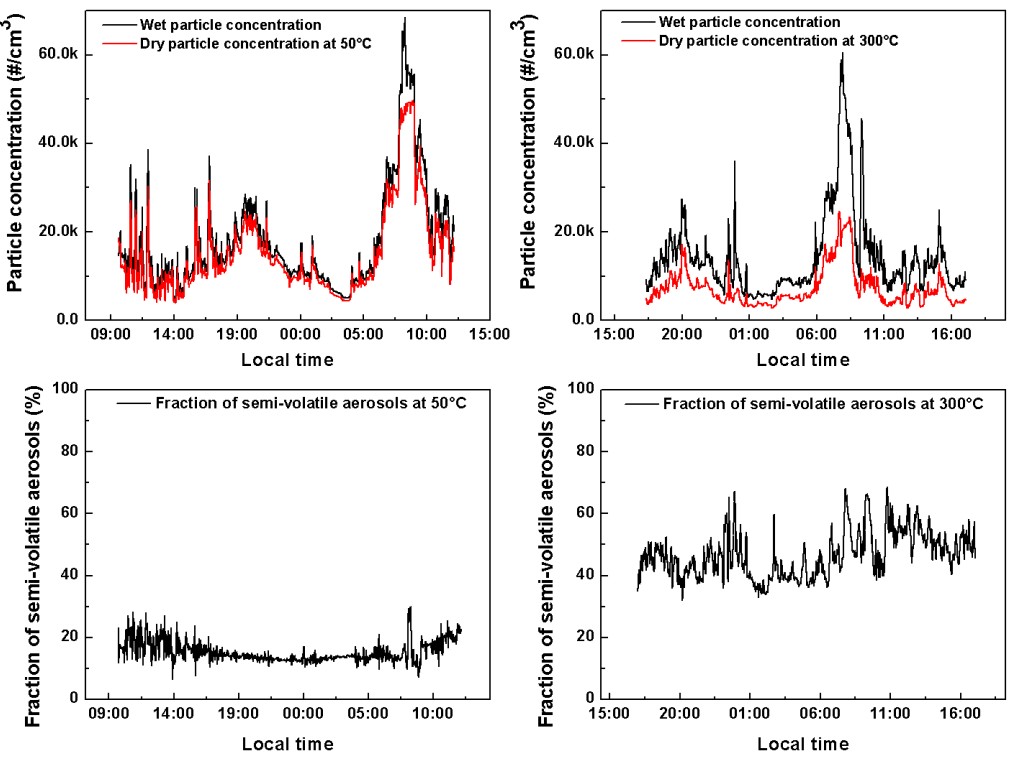


**Fig. 4.** Comparison of diurnal variation of particle number concentration of wet and dry aerosol at TDD set
temperatures 50°C **(a)** and 300°C **(b)**. Diurnal variation of fraction of dry particles at TDD set temperatures 50°C **(c)**
and 300°C **(d)**.








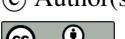




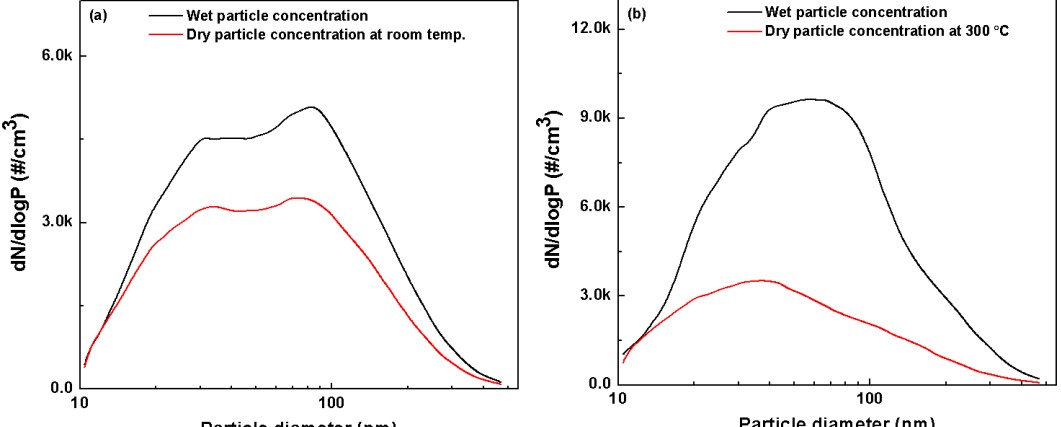


**Fig. 5.** Particle size distribution of wet and dry aerosol at room temperature **(a)** and 300°C **(b)**.

















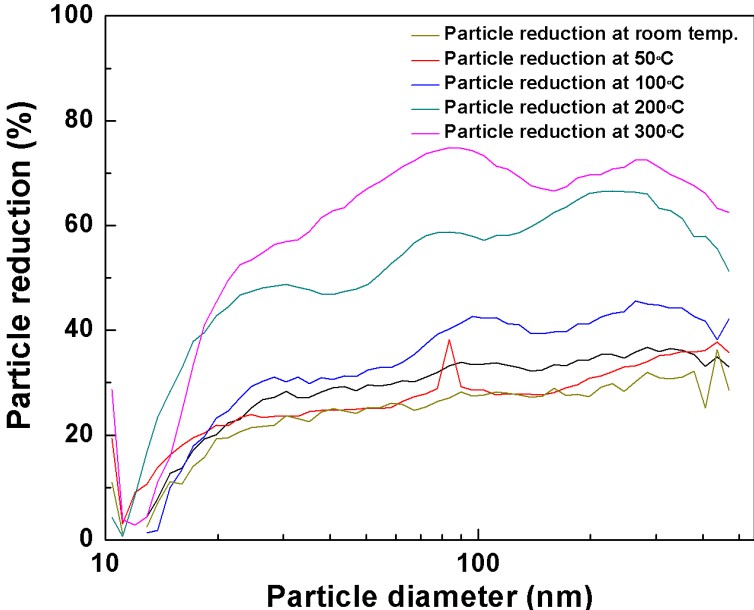


**Fig. 6.** Percentage of particle reduction from wet aerosol in their respective size bins at different TDD set

temperatures.




**Fig. 7.** Comparison of dry and wet BC absorption at TDD set temperatures- 50°C and 300°C at wavelengths- 880nm
**(a)** and 370nm **(b)**. **(c)** Boxplot of loss of absorption at different TDD set temperatures of 880nm and 370nm
wavelengths.



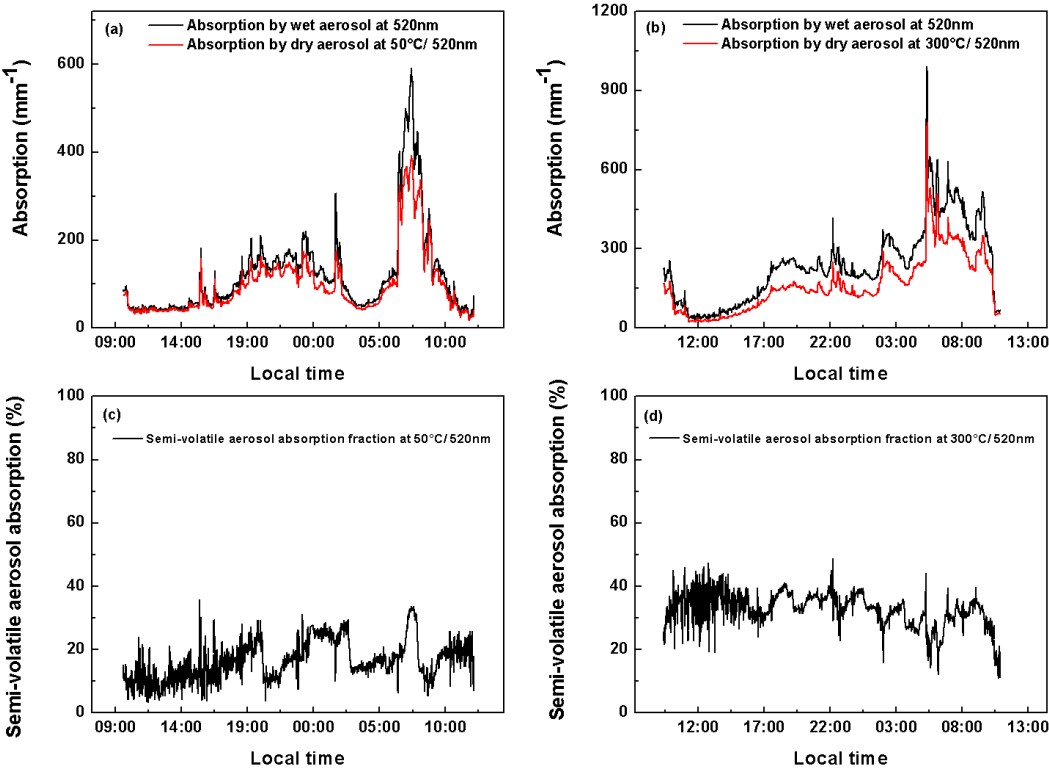


**Fig. 8.** Comparison of wet and dry aerosol absorption diurnal variation at 520nm wavelength for TDD set temperatures
of 50°C **(a)** and 300°C **(b)**. Diurnal variation of semi-volatile aerosol absorption fraction at 520 nm wavelength for
TDD set temperatures 50°C **(c)** and 300°C **(d)**.





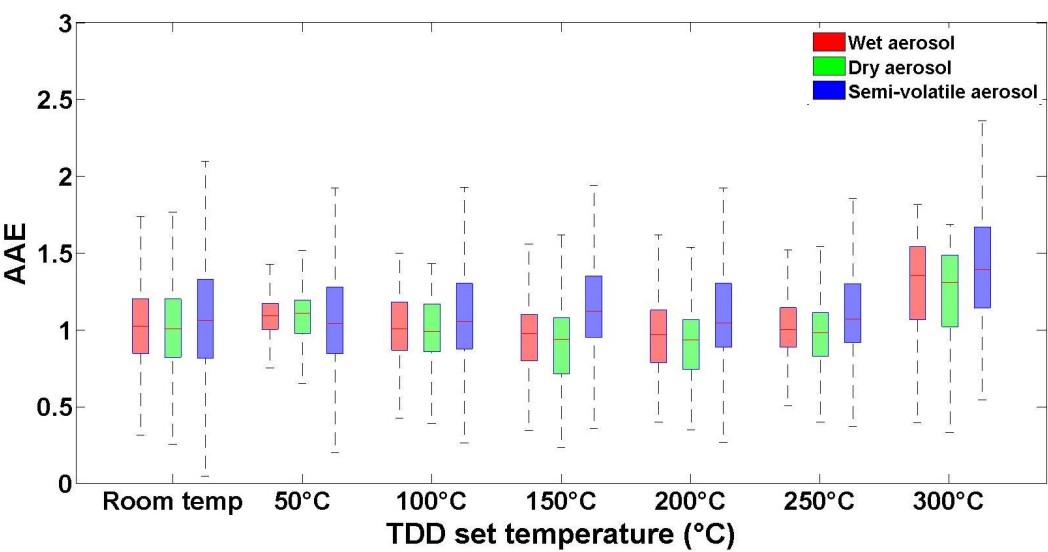

**Fig. 9.** Boxplot of Absorption Angstrom Exponent (AAE) of wet, dry and semi-volatile aerosol at different TDD set temperatures. AAE values were computed over the range of 370nm-970nm wavelengths.






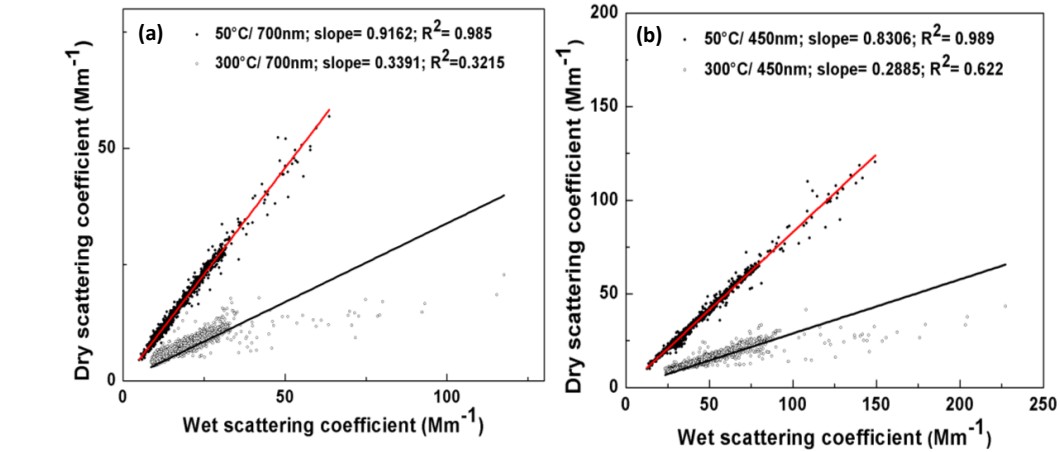

**Fig. 10.** Comparison of wet versus dry scattering coefficient TDD set temperatures- 50°C and 300°C at wavelength
700nm **(a)** and 450nm **(b)**.





**Fig. 11. (a)** Boxplot of scattering loss at different TDD set temperatures at 450nm, 550nm and 700nm wavelengths.

**(b)** Boxplot of Scattering Angstrom Exponent (SAE) of wet, dry and semi-volatile aerosol at different TDD set

temperatures. SAE was computed over the range of 450nm-700nm wavelengths.