# Peer review of "Influence of semi-volatile aerosols on physical and optical"

_Atmospheric Chemistry and Physics, 2017_

## Short Comment (SC1) · 29 Apr 2017

Authors did not talk about the drying of ambient aerosol which would bias the aerosol optical properties measurements due to hygroscopic nature of ambient aerosol. Since typical relative humidity (RH) at Kathmandu would be 60 % or higher, this will have a large impact on scattering measurements. There is evidence that on changing RH from 40 to 90 % scattering changes by the factor of 1.6 at the green wavelength (Arnott et al., 2003). Were the data corrected for RH?

There will be particle loss in thermal denuder due to thermophoretic and diffusional processes (Wehner et al., 2002). In addition, activated carbon used in the cooling section also introduce additional particle losses. Past studies show 10-30 % of particles

losses in Thermal denuder with activated carbon in cooling section depending on flow, TD configurations, and the set point temperature (Pokhrel et al., 2017; Fierz et al., 2007; Wehner et al., 2002). This loss will impact particles number concentration as well as optical measurements. How authors account this effect in their study?

References: Arnott, W. P., H. MoosmuÂÍller, P. J. Sheridan, J. A. Ogren, R. Raspet, W. V. Slaton, J. L. Hand, S. M. Kreidenweis, and J. L. Collett Jr., Photoacoustic and filter-based ambient aerosol light absorption measurements: Instrument comparisons and the role of relative humidity, J. Geophys. Res., 108(D1), 4034, doi:10.1029/2002JD002165, 2003 Fierz, M., Vernooij, M. G. C. and Burtscher, H.: An improved low-flow thermodenuder, J. Aerosol Sci., 38(11), 1163–1168, 2007. Pokhrel, R. P., Beamesderfer, E. R., Wagner, N. L., Langridge, J. M., Lack, D. A., Jayarathne, T., Stone, E. A., Stockwell, C. E., Yokelson, R. J., and Murphy, S. M.: Relative importance of black carbon, brown carbon, and absorption enhancement from clear coatings in biomass burning emissions, Atmos. Chem. Phys., 17, 5063-5078, doi:10.5194/acp-17-5063-2017, 2017. Wehner, B., Philippin, S., Wiedensohler, A.: Design and calibration of a thermodenuder with an improved heating unit to measure the size-dependent volatile fraction of aerosol particles, Journal of Aerosol Science, 33, 1087-1093, 2002.

---

## Referee Comment (RC1) · Anonymous Referee #1 · 15 Jun 2017

The authors reported a good quality results of the important problem in the Kathmandu Valley, the did not defined any where in the paper about Semi-volatile aerosol they are targeting, what are the possible sources and what are the pollutants concentrations in Kathmandu valley. In figure it has been shown that absorption of dry and wet aerosol species is maximum in the morning time from 05:00 to 10:00 Hrs, while the absorption at during this time at 50 C and 300 C is less, what is the possible reason , need some explanation,

The ambient meteorological conditions of the study period has not been defined , need to discuss

Table 3 shows that with increase in temperature loss of absorption at both wavelength (370nm and 880nm) shows similar linear trend, does it indicate something about composition of semi-volatile aerosol.

The english used in paper should be improved, many times it has been written We did, we computed etc it should be avoided and corrected.
* * *

---

## Referee Comment (RC2) · Anonymous Referee #2 · 5 Jul 2017

Review

General

The manuscript describes measurements of aerosol number concentrations, number size distributions, aerosol optical properties and volatility in Kathmandu Valley, Nepal. The air in the valley is very polluted as is shown by the referenced papers. The instrumentation used in the work could in principle have brought interesting new information on the aerosol physical properties in the region but there are too many strange things in the manuscript that it would bring any relevant and reliable information.

You explain in section 4.4, L367 "The constraint of the experiment is that the TDD flow

rate, which is restricted to 3 Lpm. Because of this we could not connect multiple instruments to maintain the TDD flow rate". This should have been mentioned much earlier, in the methodology section. And still, you could have split the 3 LPM to two instruments, for instance the CPC and the SMPS which would have made the results a bit more meaningful. Also the neph and the aethalometer could have been used at lower flow rates and then you would have had simultaneous scattering and absorption data. Further: why did you not run all the instruments on the non-heated branch of the inlet, it should not have any flow restrictions? Then you would at least have continuous time series of the relevant parameters in ambient conditions which would have incereased the value of the work. But that is speculation, now the data are here.

My criticism can be found in the detailed comments below.

Detailed comments

1) How large is the the amount of data? It is written on L102 "a few days" . This not good enough

2) How did you calibrate the nephelometer?

3) There is no description of the inlet. Cutoff diameter?

4) The length of the cycles varied: for some the temperature was kept constant during 24 hours, for some 1 hour. The results are not comparable because aerosol composition definitely varies within a day and also days are different.

5) In the manuscript "dry sample" refers to measurements carried out with instruments coupled with TDD. Well, if you heat it to 50'C or more, it is dry. But if the denuder is at room temp the sample air is just as humid as the one without the TDD.

6) L317 it is written "If we assume BC mixing state absorption affects is similar at 370nm and 880nm wavelengths, then the difference between 370nm to 880nm wavelength absorption is contributed by brown carbon" Well, the effect of coating depends on wavelength so definitely at wavelenghts so far from each other the effect is very

different. It is not possible to calculate the contribution of BrC this way.

7) L346 - 348 it is written: "Dry absorption was deducted from wet absorption values to compute the semi-volatile aerosol absorption. Then semi volatile aerosol absorption values were used to compute AAE for the semi-volatile fraction. We computed AAE over the range of wavelengths 370nm-970nm for wet, dry and semi-volatile aerosols individually". From the wet absorption, How can you deduce dry absorption at different temperatures?? Deduction: the process of reaching a decision or answer by thinking about the known facts. Linked to this, in Table 3 there are the AAE values of wet, dry and semi-volatile aerosols at different temperatures of the thermodenuder. What is this: wet aerosol at the temperature of 300°C? It is not wet then anymore. The same applies to Table 4.

8) The scattering Ångström exponent (SAE) of the semivolatile aerosol in Table 4 reaches values like $5.5 \pm 2.0$. This is another indication that there is something seriously wrong with the method. The authors obviously do not know that SAE depends very strongly on particle size and for the smallest realistic light scatterers in the atmosphere, the gas molecules, SAE = 4. It is called Rayleigh scattering.

9) The way single-scattering albedo is calculated and discussed (section 4.5) makes no sense. The nephelometer and the aethalometer were used in different days. And then you derive somehow the SSA of the semivolatile aerosol fraction at different temperatures. Even when both an aethalometer and a nephelometer are used correctly connected simultaneously to an inlet, SSA has high uncertainties. And even then, after heating sample air and volatilizing the shell of particles you would be able to get the SSA of the core but not that of the semivolatile particles. The semivolatile material is very probably as a shell on an absorbing core, not as externally mixed semivolatile particles. By heating the particle also its size changes which then changes the whole optics.

---

## Author Comment (AC1) · 27 Aug 2017

**The point by point response to the short comment**

**1. Authors did not talk about the drying of ambient aerosol which would bias the aerosol optical properties measurements due to hygroscopic nature of ambient aerosol.**
**Response:** The first set of experiments in this study accounted contribution of the semi-volatile aerosols to the total aerosol number concentration. We clearly observe a significant loss in aerosol number at each TDD set temperature (detailed summary in Table 2 of the manuscript). If only hygroscopic growth dominated aerosol optical properties, then a significant particle loss at each temperature would not have occurred. Hence it is appropriate to refer to optical properties due to loss of the semi-volatile aerosol.

**2. Since typical relative humidity (RH) at Kathmandu would be 60 % or higher, this will have a large impact on scattering measurements. There is evidence that on changing RH from 40 to 90 % scattering changes by the factor of 1.6 at the green wavelength (Arnott et al., 2003). Were the data corrected for RH?**
**Response:** For the scattering study, TSI integrating nephelometer 3563 was used. This nephelometer uses an internal 75 Watts DC halogen lamp. The heat created by this lamp in the measurement column doesn't allow the internal RH to reach as high as ambient conditions. Further, during the entire experiment, the average difference in RH values between wet and dry nephelometer was below 4% (scatter plot below). The statistical analysis shows both the instruments did not have significant RH differences over the entire measurement duration. The slope was near to 1 and thus, compared to the particle loss, the RH difference within both the instruments was not significant.

[Figure]

**Fig: Correlation plot between RH in wet and dry nephelometer measured during entire sampling duration.**

**3. There will be particle loss in thermal denuder due to thermophoretic and diffusional processes (Wehner et al., 2002). In addition, activated carbon used in the cooling section also introduce additional particle losses. Past studies show 10-30 % of particles losses in Thermal denuder with activated carbon in cooling section depending on flow, TD configurations, and the set point temperature (Pokhrel et al., 2017; Fierz et al., 2007; Wehner et al., 2002). This loss will impact particles number concentration as well as optical measurements. How authors account this effect in their study?**

**Response:** As mentioned in the manuscript (Line 206-213), TDD was operated at room temperatures to account for the particle losses due to thermophoretic and diffusional losses. We report a 12% particle loss when TDD was operated at room temperature. However, it is not possible to quantify particle loss due to TDD column and semi-volatile aerosols at higher TDD set temperatures at the same time. Hence, authors believe that this problem was properly addressed and do not require further investigation. Proper references have also been presented in the manuscript regarding use of similar thermal analytical methods for studying volatility and composition of ambient aerosol (Ishizaka & Adhikari, 2003; Murugavel & Chate, 2011). Similar experiments with TDD set at room temperature were conducted with number concentration, size distribution, absorption and scattering to quantify the respective losses due to TDD column.

---

## Author Comment (AC2) · 27 Aug 2017

**Response to comments of Reviewer 1**

**1. The authors reported a good quality results of the important problem in the Kathmandu Valley, the did not defined anywhere in the paper about Semi-volatile aerosol they are targeting, what are the possible sources and what are the pollutants concentrations in Kathmandu valley.**

**Response:** The definition was existing in the original manuscript on line no. 47-54. We modified the paragraph on the Kathmandu valley air quality situation and emission sources, which can be seen on line no. 74-91 of the modified manuscript and the same text is quoted below.

"The Kathmandu Valley in Nepal is a polluted area in South Asia and generates much interest due to its rapid urbanization, emission sources, topography and proximity to the Himalayas. Emissions from heavy traffic movements, brick-kilns, open burning of solid waste, as well as from households and industrial activities in particular are primary sources of pollution within the valley. Several studies have been conducted in the Kathmandu valley to quantify the mass of black carbon (BC), $PM_{2.5}$ and $PM_{10}$ and to identify the seasonality of air pollution (Aryal et al., 2009; Majumder et al., 2012; Putero et al., 2015; Sharma et al., 2012). These studies showed the highest aerosol loading occurs during winter season: BC in winter was ~14 µg/m$^3$ (Sharma et al., 2012) while $PM_{10}$ reached ~320 µg/m$^3$ (Putero et al., 2015). This is due primarily to strong inversion and calm weather conditions during the winter months.
Studies on source apportionment of ambient air quality in the Kathmandu valley based on NMVOC's indicate that brick kilns (~10%), traffic (~17%), residential biofuel and waste disposal (~11%), and industries (32%) are the major sources of pollution (Sarkar et al., 2017). Similar findings but with slight variation in percentages were also observed when estimating sources of EC and OC from $PM_{10}$ (Kim et al., 2015), PAH's from TSP (Chen et al., 2015), $PM_{2.5}$ and bulk aerosol studies (Shakya et al., 2010, 2017). Although none of the studies quantified the contribution of the semi-volatile aerosols to the ambient atmosphere directly. Source apportionment studies of $PM_{2.5}$ and $PM_{10}$ within the valley indicate ~50% contribution from semi-volatile aerosols composed primarily of OC, $NH_4$ and $SO_4$. This fraction, however, varies from time to time depending upon the sampling period and sampling method. In this context, it important to study the impact of the semi-volatile aerosols on physical and optical properties of aerosols in the Kathmandu Valley. To the best of our knowledge, this is the first study of its kind in this area."

**2. In figure it has been shown that absorption of dry and wet aerosol species is maximum in the morning time from 05:00 to 10:00 hrs, while the absorption at during this time at 50 C and 300 C is less, what is the possible reason, need some explanation.**

**Response:** The figure in original manuscript was also showing BC peak during morning hours. As the experiment started just after 10 a.m. and the figure X-axis scale was not able to capture this properly, hence it might be misleading. The modified figure with proper X-axis scale is given below (Fig A) as well as changed in the revised manuscript (Fig 8). However, at 300°C TDD set temperature, both wet and dry BC diurnal variations does not exhibit strong diurnal cycle like other days this may be due to change in local meteorological conditions. But still peak concentrations were observed during morning hours. For further verification we are providing similar experiment results from other day (plot given below Fig B) which clearly indicates BC peaks were observed in the morning hours.

[Figure]

Fig. A. Comparison of wet and dry aerosol absorption diurnal variation at 520nm wavelength for TDD set temperatures of $50^{\circ}$C (a) and $300^{\circ}$C (b). Diurnal variation of semi-volatile aerosol absorption fraction at 520 nm wavelength for TDD set temperatures $50^{\circ}$C (c) and $300^{\circ}$C (d).

[Figure]

Fig B. Comparison of wet and dry aerosol absorption diurnal variation at 520nm wavelength for TDD set temperatures of $50^{\circ}$C (a) and $300^{\circ}$C (b). Diurnal variation of semi-volatile aerosol absorption fraction at 520 nm wavelength for TDD set temperatures $50^{\circ}$C (c) and $300^{\circ}$C (d).

**3. The ambient meteorological conditions of the study period has not been defined, need to discuss.**

**Response:** Taking into account the reviewer's suggestion, we have incorporated a separate paragraph in section 2 of the modified manuscript (Line no. 104-111) discussing the prevailing meteorological condition over the experimental site during the study period.

The text has been quoted below for your reference.

"The general meteorology in Kathmandu during the observation period (https://www.wunderground.com/) included a mean temperature of 21.8±3.1 °C and average relative humidity of 75±10.2 %. The daily average wind speed was approximately 1.3 m s$^{-1}$ indicating prevalence of light air conditions during the sampling time period. The dominant wind direction during this period was westerlies (South West-North West) and easterlies (South East-North East) due to the presence of high mountain peaks on the northern and southern fringes of the Kathmandu valley (Panday et al., 2009; Regmi and Maharjan, 2015). Atmospheric pressure was also observed to be ~868 hPa. In these weather conditions, all measurements were carried out within the span of few months (Table 1). Care was taken so that no experiments were conducted during any occasional rain events"

**4. Table 3 shows that with increase in temperature loss of absorption at both wavelength (370nm and 880nm) shows similar linear trend, does it indicate something about composition of semi-volatile aerosol.**

**Response:** The similar trend observed in loss of absorption at both 370nm and 880nm with the increase in temperature indicates uniform mixing of aerosols. This linear trend can also be attributed to intrinsic properties of the semi-volatile aerosols like refractive index, size, mixing state or brown carbon (BrC). To understand the actual contribution of this absorption of organic or in-organic aerosol fraction requires a different set of experiments. This will be the future scope of this study.

The necessary changes can be observed in section 4.3 line no. 342-344 while the text has been quoted below for your reference below.

"If we assume BC mixing state absorption effects are similar at 370nm and 880nm wavelengths, then the difference between 370nm to 880nm wavelength absorption may be attributed to changes in the intrinsic properties of the semi-volatile aerosol, the size of aerosols, mixing state or brown carbon (BrC), which is unknown at present."

**5. The English used in paper should be improved, many times it has been written We did, we computed etc it should be avoided and corrected.**

Response: The present manuscript has been thoroughly edited by a native English speaking editor who has been acknowledged in the 'Acknowledgement' section.

---

## Author Comment (AC3) · 27 Aug 2017

**Response to comments of Reviewer 2**

**General Comments**
**The manuscript describes measurements of aerosol number concentrations, number size distributions, aerosol optical properties and volatility in Kathmandu Valley, Nepal.**
**The air in the valley is very polluted as is shown by the referenced papers. The instrumentation used in the work could in principle have brought interesting new information on the aerosol physical properties in the region but there are too many strange things in the manuscript that it would bring any relevant and reliable information.**
**You explain in section 4.4, L367 "The constraint of the experiment is that the TDD flow rate, which is restricted to 3 Lpm. Because of this we could not connect multiple instruments to maintain the TDD flow rate". This should have been mentioned much earlier, in the methodology section. And still, you could have split the 3 LPM to two instruments, for instance the CPC and the SMPS which would have made the results a bit more meaningful. Also the neph and the aethalometer could have been used at lower flow rates and then you would have had simultaneous scattering and absorption data.**

**Response:**

We agree with the reviewer's suggestions, but the primary objective of this study was to understand the contribution of semi-volatile fraction to ambient aerosol physical and optical properties. Initially authors set an experiment with all instruments measuring wet and dry aerosols but experiment was not able to succeed due to the constraints of TDD flow and nephelometer flow system. The blower system in the nephelometer rendered us unable to set a fixed flow rate for this kind of experiment and complicated the use of several instruments connected to single sampling line. Unlike pump with a flow sensor, blower system does not give values on air flow and also if there is no free air flow, blower system may not work efficiently. Considering these facts the authors decided to conduct individual experiments to avoid any minor biases.

As suggested by the reviewer, we have shifted the sentence into the methodology section (Line no. 127) of the modified manuscript.

**Further: why did you not run all the instruments on the non-heated branch of the inlet, it should not have any flow restrictions? Then you would at least have continuous time series of the relevant parameters in ambient conditions which would have increased the value of the work. But that is speculation, now the data are here.**
We agree with the reviewer comment and this could be an area for future study. The area of research on semi-volatile aerosols in the region is extremely sparse. The present study gives some understanding on semi-volatile aerosol properties and also presents scope for future experiments.

**Detailed comments**
**1) How large is the the amount of data? It is written on L102 "a few days". This not good enough.**
**Response:** The total duration of sample collection was around 1470 hours. This includes around 490 hours of each CPC and aethalometer experiment, 470 hours of nephelometer experiment and 20 hours of SMPS experiment. As no diurnal variation was observed in the fraction of semi-volatile aerosol number, we limited the SMPS study for a few hours, which was done just to understand the variation in size distribution of semi-volatile aerosol fraction.
We have modified the text appropriately in the experimental set up section (Line no. 114-116) of the modified manuscript.

**2) How did you calibrate the nephelometer?**
**Response:** The nephelometer as well as other instruments used in this experiment were brand new ones and were used for the very first time. These instruments were all factory calibrated using $CO_2$.

**3) There is no description of the inlet. Cutoff diameter?**
**Response:** Semi-volatile aerosol fraction contribution can occur through coating on fine or coarse mode particles. In order to capture broader size range particulate matter, we preferred to sample total suspended particulate matters for our experiments. Hence, we used a waterproof total suspended particulate assembly with debris screen for protection from insects or bugs at the inlet. The inlet was regularly cleaned and O-ring in the TSP inlet were lubricated. A picture of the same has been provided below. The inlet description has been given in the modified manuscript (Line no. 120-122).

[Figure]

[Figure]

Figure: TSP assembly with debris screen used for the experiment

**4) The length of the cycles varied: for some the temperature was kept constant during 24 hours, for some 1 hour. The results are not comparable because aerosol composition definitely varies within a day and also days are different.**
**Response:** The first set of experiments was conducted using CPC, where we were able to identify the semi-volatile aerosol number fraction. The CPC based experiments did not show any strong diurnal variation of the semi-volatile aerosol number fraction. Through all TDD set temperatures, semi-volatile fractions were observed to be consistent (as shown in Figures 3, 7, 10). In order to get additional information on aerosol size distribution, experiment with SMPS setup was undertaken for one hour duration at TDD set temperature only just to understand the particle loss due to semi-volatile fraction. This one hour experiment was also repeated few times to verify the consistency.

The same has been modified in section 4.2 line no. 279-283.  "Our experimental setup using SMPS was different from other instrumental setups. In first experiment using CPC, we identified that semi-volatile aerosol number fraction which did not show any strong diurnal variation. From this point, we wanted to understand particle size loss due to the semi-volatile aerosol fraction rather than diurnal variability. Hence, we operated identical SMPSs (as described in the instrument setup section), but changed the TDD set temperature every hour. We readily acknowledge that this decision also reveals a limitation of our study."

**5) In the manuscript "dry sample" refers to measurements carried out with instruments coupled with TDD. Well, if you heat it to 50'C or more, it is dry. But if the denuder is at room temp the sample air is just as humid as the one without the TDD.**
**Response:** The TDD itself is filled with activated carbon that acts like a desiccant and hence, while air passes through it at TTD set room temperature, the experiment is considered as dry. Previous research also reports that thermodenuder creates thermophoretic and diffusional losses creating dryer conditions which will lead to particle loss. Our experiment conducted at room temperature is not assumed as "wet sample" because we observed around 12% particle loss using TDD set room temperature experiment. Hence we consider TDD at room temperature experiment also as dry one.
The same has also been explained with proper references in the text (Line no. 206-213) in the modified manuscript.

**6) L317 it is written "If we assume BC mixing state absorption affects is similar at 370nm and 880nm wavelengths, then the difference between 370nm to 880nm wavelength absorption is contributed by brown carbon" Well, the effect of coating depends on wavelength so definitely at wavelenghts so far from each other the effect is very different. It is not possible to calculate the contribution of BrC this way.**
**Response:** Agreeing with the reviewer, changes in absorption at different wavelengths can be due to intrinsic properties, different size distribution of the aerosols, mixing state and partly by brown carbon, which needs further investigation and is presently out of the scope of this study. The sentence is modified in line no. 342-344. "If we assume BC mixing state absorption affects is similar at 370nm and 880nm wavelengths, then the difference between 370nm to 880nm wavelength absorption may be contributed to changes in intrinsic properties of the semi-volatile aerosol, size of aerosols, mixing state or the Brown Carbon (BrC), which is unknown at present."

**7) L346 - 348 it is written: "Dry absorption was deducted from wet absorption values to compute the semi-volatile aerosol absorption. Then semi volatile aerosol absorption values were used to compute AAE for the semi-volatile fraction. We computed AAE over the range of wavelengths 370nm-970nm for wet, dry and semi-volatile aerosols individually". From the wet absorption, How can you deduce dry absorption at different temperatures?? Deduction: the process of reaching a decision or answer by thinking about the known facts. Linked to this, in Table 3 there are the AAE values of wet, dry and semi-volatile aerosols at different temperatures of the thermodenuder. What is this: wet aerosol at the temperature of 300◦C? It is not wet then anymore. The same applies to Table 4.**

**Response:** As mentioned in the original manuscript, 'wet' sample always represents ambient aerosol and 'dry' sample represents ambient air passing through TDD. For better clarification, how we computed AAE and SAE, we provide below text as additional supplementary material.

The semi-volatile aerosol fraction contribution to ambient aerosol properties were measured through the difference between wet and dry aerosol properties.

$$SV_{AP} = WA_{AP} - DA_{AP}$$

Where,
$SV_{AP}$ = Semi-Volatile aerosol fraction contribution which can be number, scattering or absorption
$WA_{AP}$ = Wet aerosol property which is ambient aerosol number, scattering or absorption
$DA_{AP}$ = Dry aerosol property which is TDD derived aerosol number, scattering or absorption at different TDD set temperatures

For example semi-volatile aerosol fraction absorption contribution was calculated from the below formula.

$$SV_{Abs\_\lambda} = WA_{Abs\_\lambda} - DA_{Abs\_\lambda}$$

Where,
$SV_{Abs\_\lambda}$ = Semi-Volatile aerosol fraction absorption at wavelength $\lambda$
$WA_{Abs\_\lambda}$ = Wet aerosol absorption at wavelength $\lambda$
$DA_{Abs\_\lambda}$ = Dry aerosol absorption at wavelength $\lambda$

Wet and dry aerosol absorption were measured using identical aethalometers (AE-33) at seven different wavelengths. We derived semi-volatile aerosol fraction absorption at seven different wavelengths from above equation and aethalometer's (wet and dry) absorption data.

$$AE = -\frac{log\frac{E_{\lambda 1}}{E_{\lambda 2}}}{log\frac{\lambda_1}{\lambda_2}}$$

AE= Angstrom Exponent

$E_{\lambda 1}$= Absorption/Scattering/Extinction coefficient at wavelength $\lambda 1$

$E_{\lambda 2}$= Absorption/Scattering/Extinction coefficient at wavelength $\lambda 2$

From the above equation we derived wet, dry and semi-volatile aerosol fraction absorption/scattering angstrom exponent.

**Table 3.** Summary of influence of volatility on absorption at various temperatures.

| TDD set temp. in $^\circ$C | Loss of absorption at 370nm (%) | Loss of absorption at 880 nm (%) | Absorption due to intrinsic properties or BrC | Average absorption angstrom coefficient of wet aerosol * (Avg±SD) | Average absorption angstrom coefficient of dry aerosol (Avg±SD) | Average absorption angstrom coefficient of semi-volatile aerosol fraction (Avg±SD) |
|---|---|---|---|---|---|---|
| Room temp. | 16 | 16 | 0 | 1.02±0.24 | 1.01±0.24 | 1.12±0.47 |
| 50 | 23 | 20 | 3 | 1.08±0.17 | 1.08±0.18 | 1.10±0.43 |
| 100 | 19 | 18 | 1 | 1.01±0.23 | 0.98±0.23 | 1.12±0.38 |
| 150 | 25 | 21 | 4 | 0.97±0.27 | 0.92±0.30 | 1.19±0.41 |
| 200 | 31 | 27 | 4 | 0.97±0.19 | 0.92±0.18 | 1.13±0.43 |
| 250 | 35 | 28 | 7 | 1.03±0.20 | 0.99±0.22 | 1.12±0.30 |
| 300 | 37 | 28 | 9 | 1.30±0.30 | 1.24±0.30 | 1.43±0.33 |

*Average absorption Angstrom coefficient of wet aerosols (ambient aerosol) while the simultaneous dry experiment was being conducted at TDD set temperatures.

**Table 4.** Summary of influence of volatility on scattering at various temperatures.

| TDD set temp. in °C | Loss of scattering at 450nm (%) | Loss of scattering at 550nm (%) | Loss of scattering at 700nm (%) | Average scattering Angstrom coefficient of wet aerosol * (Avg±SD) | Average scattering Angstrom coefficient of dry aerosol (Avg±SD) | Average scattering Angstrom coefficient of semi-volatile aerosol fraction (Avg±SD) |
|---|---|---|---|---|---|---|
| Room temp. | 18 | 15 | 8 | 1.94±0.45 | 1.68±0.45 | 4.35±2.46 |
| 50 | 17 | 13 | 8 | 1.76±0.38 | 1.47±0.34 | 5.52±2.01 |
| 100 | 29 | 27 | 20 | 1.92±0.42 | 1.69±0.39 | 2.85±0.32 |
| 150 | 39 | 38 | 32 | 1.96±0.44 | 1.70±0.44 | 2.69±0.71 |
| 200 | 48 | 46 | 40 | 1.93±0.42 | 1.59±0.40 | 2.65±0.64 |
| 250 | 62 | 59 | 52 | 1.94±0.44 | 1.45±0.41 | 2.61±0.46 |
| 300 | 71 | 70 | 66 | 1.99±0.46 | 1.49±0.41 | 2.47±0.80 |

*Average absorption Angstrom coefficient of wet aerosols (ambient aerosol) while the simultaneous dry experiment was being conducted at TDD set temperatures.

**8) The scattering Ångström exponent (SAE) of the semivolatile aerosol in Table 4 reaches values like 5.5 ± 2.0. This is another indication that there is something seriously wrong with the method. The authors obviously do not know that SAE depends very strongly on particle size and for the smallest realistic light scatterers in the atmosphere, the gas molecules, SAE = 4. It is called Rayleigh scattering.**

Response: The scattering loss observed in Table 3 was comparable with particle loss in Table 2.

In comparison with the repeated experiments conducted for particle number and absorption (with CPC and aethalometer) experiments, scattering loss (measured by nephelometer) at higher wavelength (700nm) was observed to be less (~8%). Whereas, at lower wavelengths scattering loss was observed to be ~15% at lower TDD set temperatures. Nephelometer's first set of experiments was analyzed and it showed high semi-volatile aerosol fraction SAE at lower TDD set temperatures. We repeated the same experiments several times to cross-check the SAE values and observed a consistent trend (Representative figure given below from few experiment days). Thus we don't suspect any errors in the experimental procedures. However, this low scattering loss at 700nm, indicates the necessity for significant in-depth future studies. The scattering contribution at 700nm wavelength may be representing bigger particles and they may have less contribution of highly volatile aerosols. This and other possibilities can be an area of future study.

Regarding the calculations, the average aerosol scattering coefficients (in $Mm^{-1}$) at 450, 550 and 700nm at 50°C were observed to be 7.1E-05, 5.0E-05 and 3.1E-05 for wet (ambient) and 6.0E-05, 4.4E-05 and 3.0E-05 for dry. The averaged difference between wet and dry aerosol scattering found to be 1.1E-05, 6.1E-06 and 8.5E-07, which is nothing but semi-volatile aerosol fraction contribution at the set TDD temperature. This semi-volatile aerosol fraction scattering angstrom exponent is 5.82. Similarly, at room temperature average aerosol scattering coefficients at 450, 550 and 700nm were observed to be 4.7E-05, 3.2E-05 and 1.9E-05 for wet, 4E-05, 2.8E-05 and 1.9E-05 for dry while 7.2E-06, 4E-06 and 9.47E-07 for semi-volatile aerosol fraction contribution. These lead to subsequent scattering angstrom exponent of 4.65. After thorough cross-check of the dataset we observed consistency in high semi-volatile aerosol fraction SAE which was due to less scattering contribution at 700nm wavelength. Above explanation has been added accordingly in the manuscript at line no. (404-407).

[Figure]

Figure: Change in scattering Ångström exponent on different days of experiments

**9) The way single-scattering albedo is calculated and discussed (section 4.5) makes no sense. The nephelometer and the aethalometer were used in different days. And then you derive somehow the SSA of the semivolatile aerosol fraction at different temperatures. Even when both an aethalometer and a nephelometer are used correctly connected simultaneously to an inlet, SSA has high uncertainties. And even then, after heating sample air and volatilizing the shell of particles you would be able to get the SSA of the core but not that of the semivolatile particles. The semivolatile material is very probably as a shell on an absorbing core, not as externally mixed semivolatile particles. By heating the particle also its size changes which then changes the whole optics.**

**Response:** Our objective is to understand the semi-volatile aerosol fraction **contribution** to ambient aerosol optical properties rather than to determine semi-volatile aerosol properties. This study is trying to explain what could be the semi-volatile fraction influence on SSA.

There is a constant fraction contribution of semi-volatile aerosol physical-optical properties in our experiments (figure 3, 7 and 10). Linear regression and correlation coefficients indicated that the average absorption and scattering losses at each temperature were almost consistent for a particular TDD set temperature with very little variation in the slope. Taking this into account, the linear slopes were used to derive the semi-volatile fraction contribution for wet (ambient) aerosol absorption and scattering. Same fractions were used to understand semi-volatile aerosol fraction contribution for given wet aerosols SSA. This will give important information on the nature of semi-volatile aerosol contribution to aerosol radiative forcing. Below we explain how we have calculated SSA values.

Single scattering albedo (SSA) is defined as the ratio of scattering to total extinction due to atmospheric aerosols as suggested in the equation below.

$$SSA = \frac{Scattering}{(Scattering + Absorption)} \tag{1}$$

Assuming wet aerosol SSA = 0.9 and scattering = 100, we derived the absorption using the above equation;

$$0.9 = \frac{100}{(Absorption + 100)} \tag{2}$$

$$=> Absorption = \frac{100-90}{0.9} \tag{3}$$

So, wet aerosol absorption = 11.11

Similarly, when we consider wet aerosol SSA = 0.95 and scattering = 100, absorption = 5.2

The semi-volatile aerosol fraction contribution derived from regression slopes were used in below equations.

$Semi-volatile\ aerosol\ scattering = wet\ aerosol\ scattering *$

$$(\%\ contribution\ of\ semi-volatile) \tag{4}$$

$Semi-volatile\ aerosol\ absorption = wet\ aerosol\ absorption *$

$$(\%\ contribution\ of\ semi-volatile) \tag{5}$$

For wet aerosols scattering =100 and absorption=11.11

Semi-volatile aerosol scattering from equn. 4 = 24.58 (Table R1 Column 3, given below) (for TDD set temperature 50°C while absorption = ((11.11*17)/100)) (Table R1 Column 2, given below)

$$SSA = \frac{Semi-volatile\ aerosol\ scattering}{Semi-volatile\ aerosol\ scattering\ +\ Semi-volatile\ aerosol\ absorption} \tag{6}$$

Semi-volatile SSA at 50°C = (24.58/(24.58+2.73))

=0.861595 (Table R1 Column 4) (for TDD set temperature 50°C)

Where;

Scattering (%) = Loss of scattering at $T_i$

Absorption (%) = Loss of absorption at $T_i$

$T_i$ = TDD set temperature

Table R1: Table for wavelength interpolation

| At 450nm | TDD temp | Absorption fraction | Scattering Fraction | SSA of semi-volatile fraction assuming wet SSA=0.9 | SSA of semi-volatile fraction assuming wet SSA=0.95 |
|---|---|---|---|---|---|
| | Room Temp | 16.57 | 18 | 0.907291 | 0.954318 |
| | 50 | 24.58 | 17 | 0.861703 | 0.930072 |
| | 100 | 19.15 | 29 | 0.931707 | 0.966802 |
| | 150 | 23.05 | 39 | 0.938435 | 0.970183 |
| | 200 | 27.96 | 48 | 0.939269 | 0.970601 |
| | 250 | 30.36 | 62 | 0.948448 | 0.975169 |
| | 300 | 31.73 | 71 | 0.952738 | 0.977289 |
| At 550nm | | | | | |
| | Room Temp | 14.7 | 15 | 0.901892 | 0.951511 |
| | 50 | 23.59 | 13 | 0.832347 | 0.913776 |
| | 100 | 18.32 | 27 | 0.92996 | 0.965919 |
| | 150 | 22.02 | 38 | 0.939566 | 0.970749 |
| | 200 | 27.04 | 46 | 0.938748 | 0.97034 |
| | 250 | 29.13 | 59 | 0.948044 | 0.974969 |
| | 300 | 30.33 | 70 | 0.954112 | 0.977966 |
| At 700nm | | | | | |
| | Room Temp | 12.73 | 8 | 0.849886 | 0.923578 |
| | 50 | 20 | 8 | 0.782779 | 0.884956 |
| | 100 | 14.89 | 20 | 0.923668 | 0.962729 |
| | 150 | 18.33 | 32 | 0.940219 | 0.971075 |
| | 200 | 23.73 | 40 | 0.938218 | 0.970074 |
| | 250 | 25.64 | 52 | 0.948109 | 0.975001 |
| | 300 | 26.79 | 66 | 0.956887 | 0.979329 |

---

## Author Comment (AC4) · 27 Aug 2017

**Influence of semi-volatile aerosols on physical and optical properties of aerosols in the Kathmandu Valley**

3

**4 Sujan Shrestha1,2, Siva Praveen Puppala1, Bhupesh Adhikary1, KundanLal Shrestha2, 5 Arnico K. Panday1**

6 1International Centre for Integrated Mountain Development (ICIMOD), Khumaltar, Lalitpur, Nepal

7 2Kathmandu University, Department of Environmental Science and Engineering, Dhulikhel, Kavre, Nepal

8 *Correspondence to:* Siva Praveen Puppala (SivaPraveen.Puppala@icimod.org)

9

**10 Abstract**

11 We conducted field study during pre-monsoon season 2015 in the urban atmosphere of the Kathmandu Valley to study 12 the influence of the semi-volatile aerosol fraction on physical and optical properties of aerosols. Our experimental 13 setup consisted of a single ambient air inlet from which the flow was split into two sets of identical sampling 14 instruments. The first set connected directly with an ambient sample while the second set received the air sample 15 through a thermodenuder (TDD). Four sets of experiments were conducted to understand aerosol number, size 16 distribution, absorption, and scattering properties using. Condensation Particle Counters (CPCs), Scanning Mobility 17 Particle Sizers (SMPSs), Aethalometers (AE33) and Nephelometers, respectively. The influence of semi-volatile 18 aerosols fraction was calculated based on the difference of aerosol properties at room temperature, 50°C, 100°C, 19 150°C, 200°C, 250°C and 300°C through set TDD temperatures on ambient samples. Our results show that with 20 increasing TDD temperatures, the evaporated fraction of semi-volatile aerosols also increased. At room temperature, 21 the semi-volatile fraction of aerosol number was 12% of ambient aerosol, while at 300°C it was as high as 49%. 22 Aerosol size distribution analysis from SMPS shows that with an increase in temperature from 50°C to 300°C, the 23 peak mobility diameter of particles shifted from around 60nm to 40nm. However, no distinct change in the effective 24 diameter of the aerosol size distribution was observed with an increase in set TDD temperature. The change in size of 25 aerosols due to loss of semi-volatile component had a stronger influence ( $\sim 70\%$ ) at larger size bins when compared 26 (~20%) to smaller bins of SMPS. At 300°C, the semi-volatile aerosol fraction amplified BC absorption by 27 approximately 28% while scattering by the semi-volatile aerosol fraction contributed up to 71% of total scattering. 28 The Scattering Angstrom Exponent (SAE) of the semi-volatile aerosol fraction was found to be more sensitive at 29 lower temperatures (<100°C) than at higher temperatures. However, the Absorption Angstrom Exponent (AAE) of 30 the semi-volatile aerosol fraction did not show any significant temperature dependence.

31

Keywords: semi-volatile aerosols, aerosol number concentration, aerosol size distribution, absorption,
 scattering, black carbon, Angstrom Exponent, Kathmandu Valley.

- 34
- 35

**36 1 Introduction**

40

- 37 Aerosols are suspended particles in the air that are solid, liquid or a mixture of both states, ranging in size from a few
- 38 nanometers to several micrometers (Warneck, 2000; Zellner, 1999). Studies have shown that aerosols have profound
- 39 effects on climate (Pöschl, 2005), human health (Kampa and Castanas, 2008; Mauderly and Chow, 2008), and
- visibility of the atmosphere (Dzubay et al., 1982). Atmospheric aerosols are among the key factors that influence the 41 earth's radiative energy balance. Aerosols affect climate directly by absorption and scattering incoming solar radiation
- 42 (Haywood and Shine, 1997; Yu et al., 2006) and indirectly by acting as cloud condensation nuclei and affecting the
- 43 optical properties and life cycles of cloud (Ishizaka and Adhikari, 2003).
  - 44 Aerosols are classified as primary or secondary depending upon their origin. Primary aerosols are particles that are 45 emitted directly from the combustion of fossil fuels and biomass, such as black carbon as well as sea salts and 46 windblown mineral dust. Secondary aerosols are formed due to condensation, oxidation and chemical transformation 47 (Seinfeld and Pandis, 2006). Secondary aerosols tend to be semi-volatile in nature (Hennigan et al., 2008). The aerosol 48 components that do not condense under normal atmospheric conditions are considered as volatile aerosols while those 49 aerosols that remain in condensed phase under certain atmospheric conditions are classified as semi-volatile. Whereas, 50 the non-volatile aerosols have negligible vapor pressure and remain in condensed phase under normal atmospheric 51 conditions (Fuzzi et al., 2006), semi-volatile aerosols are believed to contribute most significantly to the toxicity of 52 particles (Stevanovic et al., 2015). The volatility of an aerosol provides an indication about its emission sources, 53 history, and chemical composition (Capes et al., 2008). The semi-volatile fraction of an aerosol largely depends on 54 the source of aerosol generation and the atmospheric conditions around the sampling site (Robinson et al., 2007). 55 56 Past field and laboratory measurements show that the volatility of aerosols is largely influenced by reaction
  - 57 temperature and precursor gases. Previous studies show that a large fraction of aerosols is highly volatile under 150°C 58 (Ishizaka and Adhikari, 2003; Murugavel and Chate, 2011 and references therein). Measurements made by Lee et al. 59 (2010) and Murugavel and Chate (2011) indicate that the semi-volatile fraction in ambient aerosol was between 50% 60 and 80% of the number concentration. Lin (2013) reported that the potential radiative effect of secondary organic 61 aerosols (SOA), which are a major fraction of semi-volatile aerosols, on climate was around one third of total aerosols. Chung & Seinfield (2002) reported that organic carbon (OC) has a global radiative forcing of -0.09 to -0.17 Wm-2 62 63 wherein 50% of the radiative forcing was contributed by SOA. The above studies show that semi-volatile aerosols 64 play a significant role in air quality and energy budget from local to global scales.
  - 65

66 Various epidemiological studies have reported the impact of semi-volatile aerosols on human health (Dalton et al., 67 2001; Ronai et al., 1994). Some of these studies find that in higher concentrations these semi-volatile aerosols can act 68 as potential carcinogens. For example, the semi-volatile polycyclic-aromatic hydrocarbons (PAHs) are not just 69 carcinogenic but also cause genetic susceptibility and oncogene activation (Ronai et al., 1994). The exposure of semi-70 volatile components, such as dioxins, induces heart disease leading towards mortality (Dalton et al., 2001). These 71 studies point toward the need for critical assessment of the semi-volatile aerosol fraction, which can lend to better 72 understanding of human health end points.

- 74 The Kathmandu Valley in Nepal is a polluted area in South Asia and generates much interest due to its rapid 75 urbanization, emission sources, topography and proximity to the Himalayas. Emissions from heavy traffic movements, 76 brick-kilns, open burning of solid waste, as well as from households and industrial activities in particular are primary 77 sources of pollution within the valley. Several studies have been conducted in the Kathmandu valley to quantify the 78 mass of black carbon (BC), PM2.5 and PM10 and to identify the seasonality of air pollution (Aryal et al., 2009; 79 Majumder et al., 2012; Putero et al., 2015; Sharma et al., 2012). These studies showed the highest aerosol loading 80 occurs during winter season: BC in winter was ~14  $\mu$ g/m3 (Sharma et al., 2012) while PM10 reached ~320  $\mu$ g/m3 81 (Putero et al., 2015). This is due primarily to strong inversion and calm weather conditions during the winter months. 82 Studies on source apportionment of ambient air quality in the Kathmandu valley based on NMVOC's indicate that 83 brick kilns (~10%), traffic (~17%), residential biofuel and waste disposal (~11%), and industries (32%) are the major 84 sources of pollution (Sarkar et al., 2017). Similar findings but with slight variation in percentages were also observed 85 when estimating sources of EC and OC from PM10 (Kim et al., 2015), PAH's from TSP (Chen et al., 2015), PM2.5 and 86 bulk aerosol studies (Shakya et al., 2010, 2017). Although none of the studies quantified the contribution of the semi-87 volatile aerosol fraction to the ambient atmosphere directly. Source apportionment studies of PM2.5 and PM10 within 88 the valley indicate ~50% contribution from semi-volatile aerosols composed primarily of OC, NH4 and SO4. This 89 fraction, however, varies from time to time depending upon the sampling period and sampling method. In this context, 90 it important to study the impact of the semi-volatile aerosols on physical and optical properties of aerosols in the 91 Kathmandu Valley. To the best of our knowledge, this is the first study of its kind in this area.
- 92
- 93

**2 Experimental site and general meteorology**

94 The Kathmandu Valley (Figure 1a) sits at 1,300m above mean sea level and is surrounded by hills as high as 2,500m 95 as shown in Figure 1b. We conducted our experiments to characterize the semi-volatile fraction of aerosols 96 contributing to the particulates of the valley. Experiments were conducted on the rooftop of the Integrated Centre for 97 International Mountain Development (ICIMOD) headquarters in Lalitpur, Nepal (27.6464° N, 85.3235° E). As shown 98 in the Figure 1b, the sampling site is located at 7 km southwest of the city center. The sampling site is primarily 99 surrounded by residential dwellings, hospitals, educational institutes, and brick kilns (the nearest brick kiln 2 km). 100 There are no obstructions around the sampling site within a 50 meter radius. During the measurement period, 101 instruments sampled ambient air from an inlet located 2m above the roof of a four story building (~15 m above the 102 ground).

103

The general meteorology in Kathmandu during the observation period (https://www.wunderground.com/) included a mean temperature of 21.8±3.1 °C and average relative humidity of 75±10.2 %. The daily average wind speed was approximately 1.3 m s-1 indicating prevalence of light air conditions during the sampling time period. The dominant wind direction during this period was westerlies (South West-North West) and easterlies (South East-North East) due to the presence of high mountain peaks on the northern and southern fringes of the Kathmandu valley (Panday et al., 2009; Regmi and Maharjan, 2015). Atmospheric pressure was also observed to be ~868 hPa. In these weather 110 conditions, all measurements were carried out within the span of few months (Table 1). Care was taken so that no
111 experiments were conducted during any occasional rain events.

112

**113 3 Experimental Setup**

114 The measurements presented in this study were made during pre-monsoon season of 2015. The total duration of 115 sample collection was approximately 1,470 hours. This collection period included approximately 490 hours of both

116 CPC and aethalometer experiments, 470 hours of nephelometer experiments, and 20 hours of SMPS experiments.

117

To use a single Thermodenuder (TDD) instrument, we arranged sets of experiments with different instrument pairs that were run sequentially. Our experimental setup consisted of an ambient air inlet split into two sets of identical sampling instruments. We used a waterproof total suspended particulate assembly with debris screen (Me One Instruments, Inc, USA) to protect the inlet from bugs and insects. The inlet was regularly cleaned and O-ring in the TSP inlet were lubricated. The first set of instruments were connected directly to an ambient sample inlet and the second set received air from the sample inlet after passing through the Thermodenuder (TDD). A schematic diagram

124 of the instrumental setup is shown in Figure 2.

125

Four sets of experiments were conducted with this set up, using four different pairs of instruments. Because the TDD flow rate is restricted to 3 Lpm, we could not connect multiple instruments to maintain the TDD flow rate. In each experiment, the TDD's temperature was changed over time to examine the fractional loss of the semi-volatile aerosol fraction at increasing temperatures. A summary of the four sets of experimental setup and respective

sampling dates are summarized in Table 1.

131

We note that the experiments had to be halted due to massive earthquake in Nepal, a period lasting from late April tolate May. In the preceding sections the term "wet" sample is used to refer to ambient measurements while the term

134 "dry" sample refers to measurements carried out with instruments coupled with TDD. TDD temperatures were set at

- room temperature, 50°C, 100°C, 150°C, 200°C, 250°C and 300°C in experiments 1, 3 and 4. Similar experimental
- temperatures were used in previous studies (Ishizaka and Adhikari, 2003; Jennings et al., 1994; Murugavel and
- temperatures were used in pre-rous statutes (ismzata and ramana), 2000, commission and ramana
- 137 Chate, 2011). In experiment 2, we examined only the relationship between particle size and volatility at room
- temperature, 50°C, 100°C, 200°C and 300°C. All the equipment used in this study were brand new and factory-
- calibrated.

140

**141 3.1 Instrument description**

142 The TDD used in this experiment is a Low-Flow (4 L min-1) Thermodenuder Model 3065, manufactured by Topas 143 GmbH, Germany (Madl et al. 2003). The TDD consists of two sections: one for desorption and the other for 144 adsorption. The TDD removes the semi-volatile fraction of an ambient sample by thermal desorption using a heating 145 element. The semi-volatile fraction that is evaporated by thermal desorption is then adsorbed by the activated carbon which is used as the working material in adsorption section. We operated the TDD only up to 300°C though the instrument has a capacity to work at temperatures up to 400°C. The activated carbon was changed regularly to ensure an optimal working state of the instrument.

149

The Condensation Particle Counters (CPC) used in this study were model 3775 manufactured by TSI Inc., USA. This instrument can detect particles as small as 4 nm in diameter and over a wide range of 0 to 107 particles per cm3. The CPC was operated with flow rate of 1.5 L min-1. Butanol was used as the working fluid and the instrument was used in the auto drain mode. Efficiency and operation of the CPC model 3775 has been further discussed by Hermann et al. (2007).

155

We operated a Scanning Mobility Particle Sizer (SMPS 3034, TSI Inc., USA) to measure the size distribution of aerosol particles. The SMPS 3034 works by separating fine particles within a range of 10 to 487 nm based on their electrical mobility. SMPS 3034 is a compact instrument with inbuilt Differential Mobility Size Analyzer (DMA) and Condensation Particle Counter (CPC). We used neutralizer model number 308701 which is x-ray based. Operation of SMPS 3034 has been further discussed by *Hogrefe et al. (2006)*.

161

162 We used a Magee Scientific Aethalometer model AE33 to study the black carbon (BC) concentration and aerosol 163 absorption in this study. During the entire measurement period the maximum attenuation limit was set at 100. The 164 instrument was operated at flow rate 2 L min-1. The Aethalometer model AE33 measures BC concentration at seven 165 different wavelengths (Drinovec et al., 2015) using filter-based light attenuation due to aerosol loading. The 166 manufacturer calibrated instrument measured light attenuation with soot particle loading. However, in real conditions 167 light attenuation reported by the instrument represents all absorbing aerosols including dust, BC, and organic carbon. 168 Light absorbing organic aerosols are also known as brown carbon (Andreae & Gelencser, 2006; Bahadur et al., 2012). 169 We followed the methodology described by Drinovec et al. (2015) to convert BC concentration to absorption by using the relationship: BC Absorption in  $mm^{-1} = BC$  (in ng  $m^{-3}$ )\* 10-3 \* MAC (the mass absorption cross sectional values). 170 171 MAC values for the specific wavelengths are given by Drinovec et al. (2015). The absorption at 880nm wavelength 172 is usually represented by BC absorption, as other particles like dust and organic carbon do not absorb or their 173 contribution to absorption is negligible at this wavelength (Singh et al., 2014). However, at the 370nm wavelength, 174 aerosol absorption is represented by all three BC, organic carbon and dust (Lim et al., 2014).

175

We employed a TSI integrating Nephelometer 3563 to measure aerosol scattering coefficient at three different wavelengths: 450nm, 550nm and 700nm. We followed the correction methodology given by *Anderson & Ogren (1998)*. A Nephelometer records both total scattering and backscattering coefficients; however, we only report results from total scattering. We operated the instruments for 24 hours at all previously reported TDD set temperatures. We followed the methodology explained by *Anderson and Ogren (1998)* to minimize angular truncation error in the

- 181 dataset using the relationship:  $\sigma_{corrected} = Correction factor (C) * \sigma_{neph}$ ; where, C is correction factor,  $\sigma_{neph}$  is the 182 scattering coefficient reported by the instrument, and  $\sigma_{corrected}$  is the corrected scattering coefficient.
- 183

**184 3.2 Quality Control**

Prior to the experiments, we conducted collocated inter comparison studies to estimate any biases within each set of identical instruments. Instruments were operated with a single inlet using a Y connector for a 24 hour period with a one minute time resolution. Thereafter, correlation between the identical instruments were calculated. The slope was approximately equal to 1 and the r2 values 0.99. No correction factors were required. (see Figure S1).

189

We conducted leakage tests on the inlet pipelines and TDD. The main inlet (Figure 2) was connected to a high efficiency particulate arrestance (HEPA) filter to verify any leakage in sampling system. HEPA filters are known to be highly efficient to produce zero aerosol concentration with a minimum 99.7% of contaminants greater than 0.3 micron (MIL-STD-282 method 102.9.1). During the beginning of each set of experiments, a HEPA filter was connected to the main inlet to check for any leakages in the setup. When the HEPA filter was connected, particle concentration readings in both the identical instruments dropped to zero as shown in Figure S2. These tests gave us confidence that the instrument setup was proper and that there were no leaks in the system.

197

**198 4 Results and discussion**

Experimental results are summarized according to aerosol number concentration, size distribution, and opticalproperties in this section.

201

**202 4.1 Influence of volatility on aerosol number concentration**

203 As discussed in Section 3, the influence of semi-volatile aerosol fraction on aerosol number concentration was 204 identified using the CPC and CPC-TDD setup. During the first experiment, the TDD operated at room temperature by 205 not providing power to TDD thermal desorption section as TDD works with only set temperatures as opposed to 206 ambient. This setup provided information about the semi-volatile aerosol fraction loss due to the dry activated carbon 207 column in the TDD. Activated carbon in the adsorption section of the TDD changes the equilibrium state of the semi-208 volatile aerosol fraction and leads to some evaporation even at room temperature (Huffman et al., 2009). We observed 209 a 12% contribution to particle loss from the semi-volatile aerosol fraction at room temperature. Similar losses of 10% 210 to 15% have also been reported in previous experiments (Fierz et al., 2007; Lee et al., 2010; Stevanovic et al., 2015). 211 These particle losses are governed by various other factors that cannot be completely avoided, such as sedimentation 212 in micro-sized particles as explained by Burtscher et al. (2001) and thermophoretic and diffusional loss in submicron 213 particles as detailed by Stevanovic et al. (2015).

- 215 The CPC and CPC-TDD setups shown in Figure 2 operated for 24 hours at each set temperature. The comparison of
- 216 wet and dry particle number concentrations at the set temperatures of 50°C and 300°C are shown in Figure 3a. The
- slope of the scatter plot gives the fraction of dry particles at a given temperature. We can see in Figure 3a that there

218 was 16% and 49% particle loss at 50°C and 300°C, respectively. A strong correlation between wet and dry CPC 219 indicates that the fraction of the semi-volatile aerosols at different ambient particle concentrations is similar. However, 220 Figure 3a also shows that the correlation between wet and dry becomes weaker when the ambient particle number 221 concentration is very high (>50000 #/cm3). Similar wet and dry particle number comparisons were obtained at the 222 other TDD set temperatures between 50°C and 300°C and these are provided in Figure 3b. Figure 3b also shows the 223 temperature dependence of the semi-volatile fraction of aerosols. Using one minute data, we calculated the particle 224 percentage loss in dry sampling, displayed in the box plots. The semi-volatile aerosol number fraction was observed 225 to be 12%, 16%, 18%, 23%, 28%, 46% and 49% at room temp., 50, 100, 150, 200, 250, 300°C set TDD temperatures 226 respectively (Table 2).

227

228 Comparing these findings with other studies, we note that Murugavel & Chate (2011) reported from Pune, India, that 229 at temperatures below 150°C, 51%-71% of the particles evaporated out of the ambient aerosol. They also reported a 230 13%-26% loss between 150-300°C and a 7%-13% loss of particles at temperatures greater than 300°C. These results 231 show that the evaporated fraction of the semi-volatile aerosols at different temperature ranges is comparatively less in 232 Kathmandu than Pune. Whereas, Murugavel & Chate (2011) used SMPS for their study, we used CPC for this study, 233 which measures particle number concentrations from 4 nm to a few microns. The SMPS used by Murugavel & Chate 234 (2011) only measures particle number concentrations between 10 nm to 487 nm. Kathmandu and Pune have different 235 source characteristics, topography, and local meteorological conditions. Thus, these differences in results with 236 Murugavel & Chate (2011) should not be surprising.

237

We compared the semi-volatile aerosol fraction from our experiment with the standard chemical compounds that have
evaporating temperatures equivalent to the TDD set temperatures. This comparison would provide vital information
about aerosol chemical composition and volatility. A similar comparison technique has been adopted by several others
studying this topic (Burtscher et al., 2001; Ishizaka & Adhikari, 2003; Murugavel & Chate, 2011).

242

243 For further analysis, we sorted aerosol volatility into two categories: (I) highly volatile for those components which 244 vaporize at temperatures  $\leq 150^{\circ}$ C, and (II) moderately volatile for those which vaporize between  $150^{\circ}$ - $300^{\circ}$ C. We found 245 23% of aerosols in our study to be highly volatile and 26% to be moderately volatile. The average aerosol number 246 concentration during our experimental period was 16,136 #/cm3. Out of this number, 2,038 #/cm3 are highly volatile 247 in nature. These particles may represent some of the highly volatile aerosol components cited in Ishizaka & Adhikari 248 (2003), such as ammonium chloride, ammonium sulphate, terpene, organic nitrogen, ethyl benzene, and sulfuric acid, 249 among others. Among the moderately volatile components cited in Ishizaka & Adhikari (2003), including diesel 250 exhaust and ammonium bisulphate, we found 6319#/cm3 aerosol particles to be moderately volatile in nature. In 251 experiments carried out by Ishizaka & Adhikari (2003) and Shrestha et al. (2014) the remaining particles which did 252 not volatilize up to 300°C were soot carbon, polymerized organic compounds, calcium carbonate, sea salt and mineral 253 dust which may represent non-volatile aerosols in our experiment. Characterizing aerosol volatility as a function of 254 chemical composition in the Kathmandu Valley needs further investigation.

256 The diurnal variation of wet and dry aerosol number concentrations and the semi-volatile aerosol fraction at 50°C and

257 300°C are shown in Figure 4. We show results from only these two temperatures to represent the minimum and

258 maximum TDD set temperatures. Results from other TDD set temperatures lie between these two temperature results.

- 259 Figure 4 (a & b) show diurnal variation in wet aerosol number concentrations which shows one peak during morning
- 260 hours around 9 a.m. and the other during evening hours around 8 p.m. Previous studies by Panday & Prinn (2009) and
- 261 Putero et al. (2015) reported a similar diurnal profile.
- 262

263 Figure 4 (c & d) shows the semi-volatile aerosol number fraction at 50°C and 300°C TDD set temperatures. The semi-264 volatile aerosol fraction did not show any diurnal variation like wet aerosols at 50°C, which indicates that the semi-265 volatile aerosol number fraction is uniform throughout the sampling period. However, as shown in Figure 4d at TDD 266 set temperature 300°C, the semi-volatile aerosol fraction fluctuates with minor spikes in wet aerosol number 267 concentration. These spikes may represent different air masses or fresh nearby sources. Thus, the semi-volatile aerosol 268 number fraction is significantly higher during peak events. However, compared to the aerosol number diurnal 269 variation, semi-volatile aerosol fraction does not exhibit strong diurnal variation (Figure 4) and the fraction remains 270 almost constant (Figure 3).

271

By comparing the diurnal variation of highly and moderately volatile aerosol fractions, we noticed highly volatile aerosol contribution was nearly consistent throughout the day while the moderately volatile aerosol fraction changed significantly during peak events (Figure S3). As mentioned above, diesel combustion is a moderately volatile aerosol source and this combustion (primarily from vehicle traffic) may account for the minor spikes in semi-volatile aerosol fraction.

277

**278 4.2 Influence of volatility on aerosol size distribution**

Our experimental setup using SMPS was different from other instrumental setups. In first experiment using CPC, we
identified that semi-volatile aerosol number fraction which did not show any strong diurnal variation. From this point,
we wanted to understand particle size loss due to the semi-volatile aerosol fraction rather than diurnal variability.
Hence, we operated identical SMPSs (as described in the instrument setup section), but changed the TDD set
temperature every hour. We readily acknowledge that this decision also reveals a limitation of our study.

284

The results of SMPS measurements are summarized in Table 2. The semi-volatile aerosol fraction was observed to be 62% and 49% during SMPS and CPC experiments respectively at TDD set temperature 300°C (see table 2). Similar behavior was seen at other temperatures as well. One possible reason for the difference between the two measurements might be that CPC reported aerosol number concentrations from 4 nm to a few microns whereas the SMPS only monitors 10nm to 487nm. The semi-volatile aerosol fraction may be higher in smaller diameter particles compared to larger diameter particles. Figure 5 shows the number size distribution for TDD at room temperature and 300°C. Even though the dry aerosol number size distribution at room temperature was significantly lower compared to wet aerosol, their peak mobility diameter did not shift significantly (from 85nm to 80nm). A similar comparison at 300°C shows a

- different result. At this temperature, the peak diameter shifted from approximately 60nm to 40nm. This correlates with
- a previous study (An et al., 2007) that reported that when individual ammonium sulphate particles of different sizes
- were evaporated at different temperatures, the particle sizes decreased significantly. Thus, we hypothesize that the
- 296 decrease in particle size of individual components of aerosol contributed in the shift in peak diameter towards smaller
- diameter as observed in our study.
- 298

299 In Figure 6, we show individual size bin semi-volatile aerosol fraction loss as a function of the particle diameter at all 300 experimental temperatures. The data show a greater reduction of aerosol size for larger diameter aerosols compared 301 to smaller diameter aerosols. This may be due to the fact that the semi-volatile aerosol fraction at larger diameters 302 may be in the form of a coating or internally-mixed state. By losing this fraction, the aerosol size is expected to 303 decrease. This conclusion is corroborated by the observation that, as the size of the aerosols decrease, the peak 304 diameter also shifts towards smaller diameters at higher temperature. The shift in peak diameter was observed to be 305 approximately 5-7 nm between wet and dry sampling at TDD set temperature  $\leq 100^{\circ}$ C, and shifted significantly to 20-306 22 nm at TDD set temperature 200°C to 300°C. Murugavel and Chate (2011) reported variable particle loss percentage 307 at different sizes and at different TTD temperatures over a yearlong study. The difference between their study and 308 ours may be attributed to differences in data collection. Murugavel and Chate (2011) reported data that represents 309 monthly and annual averages, whereas our present study represents an event sampling (one hour). Our results show 310 that the mixing process of ambient aerosols with highly and moderately volatile aerosols/precursors are different due 311 to differing emission sources and requires further investigation.

312

**313 4.3 Influence of volatility on aerosol absorption**

314 In actual atmospheric conditions, BC exists in both elemental and mixed states. An aethalometer-derived aerosol 315 absorption represents both these states. Previous studies show that non-absorbing material coatings over elemental 316 carbon (EC) enhances absorption due to the lensing effect (Lack & Cappa, 2010; Schnaiter et al., 2005; Shiraiwa et 317 al., 2010; Zhang et al., 2008). By removing this coating, elemental carbon absorption will change. Dust aerosol 318 displays absorption features in the ultraviolet (UV) through the VIS wavelengths due to its mineralogical composition; 319 however dust aerosol is non-volatile in nature. Elemental carbon can be volatilized above 600°C (Shrestha et al., 320 2014), but below 300°C it is stable. In our study, semi-volatile aerosol fraction absorption is represented either by 321 light absorbing organics or material (organic/inorganic) coated on EC.

322

Dust absorption is generally low in the spectral regime above 600 nm or tends to have a constant background absorption value for wavelengths larger than 600 nm (Gillespie and Lindberg, 1992; Lindberg et al., 1993; Sokolik and Toon, 1999; Cao et al., 2005; Kumar et al., 2008). Hence, wet and dry absorption measured by the aethalometer at 880nm wavelength is representative of either EC or mixed state of EC absorption (Lack & Cappa, 2010). Comparison of wet and dry aerosol absorption at 880nm for 50°C and 300°C is shown in Figure 7a. The semi-volatile aerosol fraction absorption contribution to the total aerosol absorption was observed to be 20% and 28% at 50°C and 329 300°C, respectively, at 880nm wavelength. Loss of absorption from 50°C to 300°C increased only 8%, which is less

- compared to particle loss of 33%. The particle loss was not proportional to the absorption loss at 880nm wavelength
- mainly due to EC and its mixed state. Since EC and dust cannot be volatilized under 300°C thus, the measured semi-
- volatile aerosol fraction absorption contribution at 880nm is mainly from EC mixed state with semi-volatile aerosol

333 fraction.

334

Highly and moderately volatile aerosol fraction contributed 21% and 7%, respectively, to aerosol absorption at 880nm wavelength. As shown in Table 3, one fourth of aerosol absorption at 880nm wavelength is contributed by the semi-volatile aerosol fraction. Wet and dry aerosol absorption at 370nm is shown in Figure 7b. The semi-volatile aerosol fraction absorption was slightly higher at 370nm compared to 880nm (Table 3). The correlation for wet and dry absorption at both wavelengths stays similar at a higher range unlike aerosol number concentrations reported by CPC in section 4.1. Results for other TDD set temperatures are summarized in Figure 7c.

341

342 If we assume BC mixing state absorption effects are similar at 370nm and 880nm wavelengths, then the difference 343 between 370nm to 880nm wavelength absorption may be attributed to changes in the intrinsic properties of the semi-344 volatile aerosol, the size of aerosols, mixing state or brown carbon (BrC), which is unknown at present. This brown 345 carbon absorption is approximately 3% and 9% at 50°C and 300°C, respectively. As shown in Table 3, highly volatile 346 aerosol fraction does not enhance aerosol absorption at lower wavelengths (0-3%). This finding indicates that highly 347 volatile aerosols are not truly representative of brown carbon aerosols. Furthermore, as our results show, the 348 moderately volatile aerosol fraction absorption does enhance 4-9% at 370nm compared to 880nm. This indicates that 349 brown carbon aerosols are moderately volatile in nature. We conclude that the brown carbon contribution is relatively 350 less (0-9%) compared to absorption enhancement (16-28%) due to EC mixing state with the semi-volatile fraction of 351 aerosol (Table 3).

352

353 In Figure 8a and 8b, we report the diurnal variation of absorption using the aethalometer and the aethalometer coupled 354 with a TDD setup at 50°C and 300°C, respectively, for 520nm wavelength. As expected, both figures show an increase 355 in BC absorption during the early morning hours, less during afternoon, and then building up again toward evening – 356 a finding similar to results in Backman et al. (2012). Figure 8c shows that at 50°C both wet and dry aerosol absorption 357 demonstrate similar magnitude as well as diurnal variation. The semi-volatile aerosol fraction absorption is 358 approximately 20% at 50°C TDD set temperature. However, as shown in Figure 8d, although the diurnal variability 359 is similar at 300°C, the semi-volatile aerosol absorption increases to nearly 30%. Unlike the CPC testing, the semi-360 volatile aerosol absorption is more variable throughout the day at TDD set temperature 50°C. It is also noteworthy 361 that the semi-volatile aerosol absorption shows similar variability at both TDD set temperatures.

362

Further, the data obtained from aethalometer were used to illuminate the wavelength dependency of absorption which is usually expressed as an Absorption Angstrom Exponent or AAE. AAE is simply the negative slope of the log of

absorption by the log of two different wavelengths. In past studies, AAE has been used to make inferences about the

dominant composition of absorbing aerosols in the atmosphere (Bergstrom et al., 2007). Several studies report AAE
values close to 1 for fossil fuel sources and close to 2 for biomass sources (Kirchstetter et al., 2004). Our results, as
summarized in Table 3 show wet aerosol AAE ranges from 0.97 to 1.30 with a median value around 1 implying
sampled aerosols are dominated by fossil fuel sources. The AAE results for 300°C shown in Figure 9 was around 1.5

- 370 indicating the influence of biomass burning source(s).
- 371

372 Dry absorption at different TDD set temperatures was deducted from simultaneous measurement of wet absorption 373 values to compute the semi-volatile aerosol fraction absorption (details given in S1). Thereafter, semi-volatile aerosol 374 fraction absorption at seven different wavelengths was used to compute AAE for the semi-volatile fraction. We 375 computed AAE over a range of wavelengths (370nm - 970nm) for wet, dry and semi-volatile aerosol fractions 376 individually (Figure 9). As also shown in Table 3, the semi-volatile aerosol fraction AAE ranged between 1.10 to 1.43. 377 The maximum semi-volatile aerosol fraction AAE value was observed when the sample was influenced by biomass 378 sources. Highly volatile aerosol AAE was recorded at 1.1 whereas moderately volatile aerosol AAE was ranged 379 between 1.1-1.4. As discussed above, the highly volatile aerosol absorption was primarily from the BC mixing state 380 and the mixing state AAE is 1.1. We compared our findings with Lack and Langridge (2013) who reported brown 381 carbon AAE ranging between 2-10. Our observed semi-volatile aerosol fraction AAE was significantly below this 382 range further corroborating the results that absorption is mainly influenced by mixing states as compared to brown 383 carbon.

384

**385 4.4 Influence of volatility on aerosol scattering**

386 In this section, we discuss the influence of volatility on the scattering properties of the aerosols. As shown in Figure 387 10a, the semi-volatile aerosol fraction scattering contribution at 700nm wavelength was observed to be 8% and 66% 388 of wet aerosol scattering at 50°C and 300°C TDD set temperatures, respectively. Whereas at 450nm wavelength, the 389 semi-volatile aerosol fraction scattering contribution increased to 17% and 71% at 50°C and 300°C TDD set 390 temperatures, respectively (Figure 10b). The influence of wavelength on scattering loss is evident for all set 391 temperatures (Figure 11a). Even though the CPC and scattering experiments were conducted on different days, by 392 assuming the urban air mass characteristics remains similar, we infer that particle loss is not proportional to scattering 393 loss (Figure 3a and Figure 10a & 10b). However the scattering loss at 700nm wavelength is somewhat similar (66% 394 to 62% versus 66% to 49%) to the particle loss in the SMPS experiment (Table 2 and Table 4). Thus, the smaller 395 particle semi-volatile aerosol fraction has greater influence in total aerosol scattering. More tests are required to 396 statistically validate these inferences. We summarize the results for all three wavelengths (450nm, 550nm and 700nm) 397 at different TDD set temperatures in Figure 11a. Diurnal variability of wet and dry aerosol scattering (figure not 398 shown) was observed similar to CPC experiment.

399

400 Using the methodology described in section 4.3 we computed the Scattering Angstrom Exponent (SAE). The semi-401 volatile aerosol fraction SAE results differ compared to AAE (Figure 11b). The semi-volatile aerosol fraction AAE

402 was similar at all TDD set temperatures while SAE shows higher values for room temperature and 50°C compared to

403 higher TDD set temperatures. The semi-volatile aerosol fraction SAE values were observed to be greater than 4 at 404 room temperature and 50°C. We repeated the same experiments several times to cross-check the SAE values and 405 observed a consistent trend. The scattering contribution at 700nm wavelength might be representing bigger particles 406 and they may have less contribution of highly volatile aerosols. However, this low scattering loss at 700nm, indicates 407 the necessity for significant in-depth future studies. This indicates that the scattering contribution of the semi-volatile 408 fraction is nearly 8 times higher at 450nm wavelength compared to the 700nm wavelength at room temperature and 409 50°C. For other TDD set temperatures, the semi-volatile aerosol fraction contribution to scattering is around three 410 times at 450nm wavelength compared to 700nm wavelength, which is slightly higher than that of wet aerosol SAE 411 (Table 4).

**412**

**413**

**4.5 Influence of volatility on aerosol single scattering albedo**

414 Single Scattering Albedo (SSA) is the ratio of scattering coefficient to extinction coefficient, which provides an 415 indication of how absorbing or scattering the sampled aerosol is. An SSA value greater than 0.95 represents aerosol 416 with a net effect of cooling whereas an SSA value less than 0.85 will have a warming effect. SSA values between 0.85 417 and 0.95 may represent warming or cooling effect depending upon surface albedo and cloud cover (Ramanathan et 418 al., 2001). We computed SSA (more details in S2) values for different semi-volatile aerosol fraction at different TDD 419 set temperatures by assuming wet aerosol SSA as 0.9 and 0.95. By assuming wet aerosol SSA at 0.9 we can derive 420 scattering and absorption coefficient values as 100X and 11X (X can be any arbitrary value). We know from sections 421 4.3 and 4.4 the amount of scattering and absorption contribution from the semi-volatile aerosol fraction. By applying 422 these fractions to the 100X and 11X values we can retrieve scattering and absorption coefficients of the semi-volatile 423 aerosol fraction and calculate the SSA. By using this method, we calculate the semi-volatile aerosol fraction SSA 424 values (Table 5).

425

426 Our results show that the semi-volatile aerosol fraction SSA was observed lower at room temperature and 50°C 427 compared to other TDD set temperature at all wavelengths. In addition, the semi-volatile aerosol fraction is more 428 absorbing at 700nm compared to the 450nm wavelength. This may due to the fact that scattering and absorption loss 429 are not similar at different wavelengths. Our results show at TDD set temperature 50°C, the semi-volatile aerosol 430 fraction SSA was observed to be minimal at all wavelengths. The semi-volatile aerosol fraction scattering loss was 431 observed relatively high compared to its absorption TDD set temperature at 50°C. This led to lower SSA values at 432 this temperature. If this process is applicable in atmospheric conditions, rising noon time temperature may influence 433 the net aerosol optical properties and make the atmosphere more absorbing in nature.

434

**435 5 Conclusion**

436 Ours is the first of its kind study to quantify the semi-volatile aerosol fraction influence on aerosol physical and optical 437 properties over the Kathmandu Valley. Experimental results show that the semi-volatile aerosol number fraction 438 ranged from 12% to 49% at TDD set temperatures from room to 300°C, respectively. During our experiment, we 439 observed that highly volatile aerosols do not exhibit diurnal variability while the contribution of moderately volatile

- 440 aerosols increases during peak concentration events. In addition, SMPS experiment results show that the reduction of 441 the aerosol size was high for larger diameter aerosols compared to smaller diameter aerosols due to removal of the 442 semi-volatile aerosol fraction. We also note that the semi-volatile aerosol fraction mixing state contributed around 443 20% to total aerosol absorption. Aerosol absorption by the semi-volatile aerosol fraction were observed to be in 444 between 16% to 28% at 880nm wavelength whereas calculated brown carbon contribution to aerosol absorption ranged 445 for a contribution to aerosol absorption ranged
- from 0 to 9%. The scattering contribution was observed to be in the range 18% to 71% and 8% to 66% at 450nm and
- 446 700nm, respectively. Our results show that the semi-volatile aerosol fraction contribution to aerosol scattering was
- significantly higher compared to aerosol absorption and number. Since the semi-volatile aerosol fraction scattering
- 448 contribution was found to be two times higher than its absorption, this implies that removal of the semi-volatile
- aerosols will lead to a more absorbent atmosphere.
- 450 In short, our study shows that the semi-volatile aerosols play important role in characterizing aerosol physical and
- 451 optical properties over the Kathmandu Valley which will further aid in understanding health and climate impacts of
- 452 aerosols. The results discussed are based on limited and unique aerosol sampling in the Kathmandu Valley and can be
- 453 improved to better characterize aerosols in the valley.
- 454

**455 Acknowledgements and Disclaimer**

- 456 This project was partially funded by core funds of ICIMOD contributed by the government of Afghanistan, Australia,
- 457 Austria, Bangladesh, Bhutan, China, India, Myanmar, Nepal, Norway, Pakistan, Switzerland, and the United
- 458 Kingdom. We thank Pradeep Dangol for his technical support during initial instrument setup. The first author is also
- 459 grateful to the Department of Environmental Science and Engineering, Kathmandu University for encouragement in
- 460 carrying out this study. The views and interpretations in this publication are those of the authors and are not necessarily
- 461 attributable to ICIMOD. We would like to acknowledge Mr. Parth Sarathi Mahapatra for scientific editing of the
- 462 manuscript. Acknowledgements are also due to Dr Christopher Butler for his English editing of the manuscript. We
- would also like to acknowledge the help of anonymous reviewers for helping us improve the quality of the manuscriptand handling editor for smooth handling of the manuscript.
- 465

**466 References**

- 467 An, W. J., Pathak, R. K., Lee, B. and Pandis, S. N.: Aerosol volatility measurement using an improved
- thermodenuder: Application to secondary organic aerosol, J. Aerosol Sci., 38(3), 305–314,
- doi:10.1016/j.jaerosci.2006.12.002, 2007.
- Anderson, T. L. and Ogren, J. a.: Determining Aerosol Radiative Properties Using the TSI 3563 Integrating
  Nephelometer, Aerosol Sci. Technol., 29(1), 57–69, doi:10.1080/02786829808965551, 1998.
- 472 Andreae, M. O. and Gelencsér, A.: Black carbon or brown carbon? The nature of light-absorbing carbonaceous
  473 aerosols, Atmos. Chem. Phys., 6(3), 3419–3463, doi:10.5194/acpd-6-3419-2006, 2006.
- Aryal, R. K., Lee, B. K., Karki, R., Gurung, A., Baral, B. and Byeon, S. H.: Dynamics of PM2.5 concentrations in
  Kathmandu Valley, Nepal, J. Hazard. Mater., 168(2–3), 732–738, doi:10.1016/j.jhazmat.2009.02.086, 2009.
- Backman, J., Rizzo, L. V., Hakala, J., Nieminen, T., Manninen, H. E., Morais, F., Aalto, P. P., Siivola, E., Carbone,
  S., Hillamo, R., Artaxo, P., Virkkula, A., Petäjä, T. and Kulmala, M.: On the diurnal cycle of urban aerosols, black

- 478 carbon and the occurrence of new particle formation events in springtime São Paulo, Brazil, Atmos. Chem. Phys.,
  479 12(23), 11733–11751, doi:10.5194/acp-12-11733-2012, 2012.
- 480 Bahadur, R., Praveen, P. S., Xu, Y. and Ramanathan, V.: Solar absorption by elemental and brown carbon
- determined from spectral observations., Proc. Natl. Acad. Sci. U. S. A., 109(43), 17366–71,
- doi:10.1073/pnas.1205910109, 2012.
- 483 Bergstrom, R. W., Pilewskie, P., Russell, P. B., Redemann, J., Bond, T. C., Quinn, P. K., Sierau, B., Physics, S.,
- 484 Sciences, O., Marine, P., Oceanic, N. and Science, C.: Spectral absorption properties of atmospheric aerosols,
   485 Atmos. Chem. Phys., 5937–5943, 2007.
- Burtscher, H., Baltensperger, U., Bukowiecki, N., Cohn, P., Hu, C., Mohr, M., Matter, U., Nyeki, S., Schmatloch,
  V., Streit, N. and Weingartner, E.: Separation of volatile and non-volatile aerosol fractions by thermodesorption :
- **488** instrumental development and applications, , 32, 427–442, 2001.
- 489 Cao, J. J., Lee, S. C., Zhang, X. Y., Chow, J. C., An, Z. S., Ho, K. F., Watson, J. G., Fung, K., Wang, Y. Q. and
- 490 Shen, Z. X.: Characterization of airborne carbonate over a site near Asian dust source regions during spring 2002
  491 and its climatic and environmental significance, J. Geophys. Res. D Atmos., 110(3), 1–8,
- **492** doi:10.1029/2004JD005244, 2005.
- 493 Capes, G., Johnson, B., McFiggans, G., Williams, P. I., Haywood, J. and Coe, H.: Aging of biomass burning
- 494 aerosols over West Africa: Aircraft measurements of chemical composition, microphysical properties, and emission
   495 ratios, J. Geophys. Res., 113(September), D00C15, doi:10.1029/2008JD009845, 2008.
- 496 Chen, P., Kang, S., Li, C., Rupakheti, M., Yan, F., Li, Q., Ji, Z., Zhang, Q., Luo, W. and Sillanpää, M.:
- 497 Characteristics and sources of polycyclic aromatic hydrocarbons in atmospheric aerosols in the Kathmandu Valley,
  498 Nepal, Sci. Total Environ., 538(AUGUST 2015), 86–92, doi:10.1016/j.scitotenv.2015.08.006, 2015.
- Chung, S. H. and Seinfeld, J. H.: Global distribution and climate forcing of carbonaceous aerosols, J. Geophys. Res.
  Atmos., 107(19), doi:10.1029/2001JD001397, 2002.
- 501 Cross, E. S., Onasch, T. B., Ahern, A., Wrobel, W., Slowik, J. G., Olfert, J., Lack, D. a., Massoli, P., Cappa, C. D.,
- 502 Schwarz, J. P., Spackman, J. R., Fahey, D. W., Sedlacek, A., Trimborn, A., Jayne, J. T., Freedman, A., Williams, L.
- 503 R., Ng, N. L., Mazzoleni, C., Dubey, M., Brem, B., Kok, G., Subramanian, R., Freitag, S., Clarke, A., Thornhill, D.,
- Marr, L. C., Kolb, C. E., Worsnop, D. R. and Davidovits, P.: Soot Particle Studies—Instrument Inter-Comparison—
   Project Overview, Aerosol Sci. Technol., 44(8), 592–611, doi:10.1080/02786826.2010.482113, 2010.
- $505 \qquad 110jcct Overview, Acrosof Sci. 1ccinitor., 44(6), 592-011, doi:10.1080/02780820.2010.462115, 2010.$
- 506 Dalton, T. P., Kerzee, J. K., Wang, B., Miller, M., Dieter, M. Z., Lorenz, J. N., Shertzer, H. G., Nebert, D. W. and
- Puga, A.: Dioxin Exposure Is an Environmental Risk Factor for Ischemic Heart Disease, Cardiovasc. Toxicol., 1(4),
  285–298, doi:10.1385/CT:1:4:285, 2001.
- 509 Drinovec, L., Močnik, G., Zotter, P., Prévôt, A. S. H., Ruckstuhl, C., Coz, E., Rupakheti, M., Sciare, J., Müller, T.,
- 510 Wiedensohler, A. and Hansen, A. D. A.: The "dual-spot" Aethalometer: an improved measurement of aerosol black
- 511 carbon with real-time loading compensation, Atmos. Meas. Tech., 8(5), 1965–1979, doi:10.5194/amt-8-1965-2015,
  512 2015.
- 513 Dzubay, T. G., Stevens, R. K., Lewis, C. W., Hern, D. H., Courtney, W. J., Tesch, J. W. and Mason, M. A.:
- 514 Visibility and aerosol composition in Houston, Texas, Environ. Sci. Technol., 16(8), 514–525,
- 515 doi:10.1021/es00102a017, 1982.
- Fierz, M., Vernooij, M. G. C. and Burtscher, H.: An improved low-flow thermodenuder, J. Aerosol Sci., 38(11),
  1163–1168, doi:10.1016/j.jaerosci.2007.08.006, 2007.
- 518 Fuzzi, S., Andreae, M. O., Huebert, B. J., Kulmala, M., Bond, T. C., Boy, M., Doherty, S. J., Guenther, A. and
- 519 Nazionale, C.: and Physics Critical assessment of the current state of scientific knowledge, terminology, and
- research needs concerning the role of organic aerosols in the atmosphere , climate , and global change, Atmos.
- **521** Chem. Phys., 6, 2017–2038, 2006.
- Gillespie, J. B. and Lindberg, J. D.: Ultraviolet and visible imaginary refractive index of strongly absorbing
   atmospheric particulate matter, Appl. Opt., 3–6, doi:10.1364/AO.31.002112, 1992.
- 524 Haywood, J. M. and Shine, K. P.: Multi-spectral calculations of the direct radiative forcing of tropospheric sulphate
- and soot aerosols using a column model, Q. J. R. Meteorol. Soc., 123(543), 1907–1930,
- 526 doi:10.1002/qj.49712354307, 1997.

- 527 Hennigan, C. J., Sullivan, A. P., Fountoukis, C. I., Nenes, A., Hecobian, A., Vargas, O. and Peltier, R. E.: and
- Physics On the volatility and production mechanisms of newly formed nitrate and water soluble organic aerosol in
   Mexico City, , (x), 3761–3768, 2008.
- 530 Hermann, M., Wehner, B., Bischof, O., Han, H.-S., Krinke, T., Liu, W., Zerrath, A. and Wiedensohler, A.: Particle
- 531 counting efficiencies of new TSI condensation particle counters, J. Aerosol Sci., 38(6), 674–682,
- **532** doi:10.1016/j.jaerosci.2007.05.001, 2007.
- 533 Hogrefe, O., Lala, G. G., Frank, B. P., Schwab, J. J. and Demerjian, K. L.: Field Evaluation of a TSI Model 3034
- Scanning Mobility Particle Sizer in New York City: Winter 2004 Intensive Campaign, Aerosol Sci. Technol.,
  40(10), 753–762, doi:10.1080/02786820600721846, 2006.
- Huffman, J. A., Docherty, K. S., Aiken, A. C., Cubison, M. J., Ulbrich, I. M., Decarlo, P. F. and Sueper, D.: and
  Physics Chemically-resolved aerosol volatility measurements from two megacity field studies, 7161–7182, 2009.
- Ishizaka, Y. and Adhikari, M.: Composition of cloud condensation nuclei, J. Geophys. Res., 108(D4), 4138,
  doi:10.1029/2002JD002085, 2003.
- Jennings, S. G., O'Dowd, C. D., Cooke, W. F., Sheridan, P. J. and Cachier, H.: Volatility of elemental carbon,
  Geophys. Res. Lett., 21(16), 1719–1722, doi:10.1029/94GL01423, 1994.
- 542 Kampa, M. and Castanas, E.: Human health effects of air pollution, Environ. Pollut., 151(2), 362–367,
- 543 doi:10.1016/j.envpol.2007.06.012, 2008.
- 544 Kim, B. M., Park, J.-S. S., Kim, S.-W. W., Kim, H., Jeon, H., Cho, C., Kim, J.-H. H., Hong, S., Rupakheti, M.,
- Panday, A. K., Park, R. J., Hong, J. and Yoon, S.-C. C.: Source apportionment of PM10 mass and particulate carbon in the Kathmandu Valley, Nepal, Atmos. Environ., 123(November), 190–199, doi:10.1016/j.atmosenv.2015.10.082, 2015.
- 548 Kirchstetter, T. W., Novakov, T. and Hobbs, P. V.: Evidence that the spectral dependence of light absorption by 549 aerosols is affected by organic carbon, J. Geophys. Res., 109(D21), D21208, doi:10.1029/2004JD004999, 2004.
- Kumar, S., National, M. and Kanpur, T.: Modeling optical properties of mineral dust over the Indian Desert, ,
   (December), doi:10.1029/2008JD010048, 2008.
- Lack, D. a. and Cappa, C. D.: Impact of brown and clear carbon on light absorption enhancement, single scatter
- albedo and absorption wavelength dependence of black carbon, Atmos. Chem. Phys., 10(9), 4207–4220,
  doi:10.5194/acp-10-4207-2010, 2010.
- Lack, D. A. and Langridge, J. M.: On the attribution of black and brown carbon light absorption using the Ångström
  exponent, Atmos. Chem. Phys., 13(20), 10535–10543, doi:10.5194/acp-13-10535-2013, 2013.
- 557 Lee, B. H., Kostenidou, E., Hildebrandt, L., Riipinen, I., Engelhart, G. J., Mohr, C., DeCarlo, P. F., Mihalopoulos,
- 558 N., Prevot, a. S. H., Baltensperger, U. and Pandis, S. N.: Measurement of the ambient organic aerosol volatility
- distribution: application during the Finokalia Aerosol Measurement Experiment (FAME-2008), Atmos. Chem.
  Phys., 10(24), 12149–12160, doi:10.5194/acp-10-12149-2010, 2010.
- Lim, S., Lee, M., Kim, S. W., Yoon, S. C., Lee, G. and Lee, Y. J.: Absorption and scattering properties of organic
  carbon versus sulfate dominant aerosols at Gosan climate observatory in Northeast Asia, Atmos. Chem. Phys.,
  14(15), 7781–7793, doi:10.5194/acp-14-7781-2014, 2014.
- Lin, G.: Global modeling of secondary organic aerosol formation: from atmospheric chemistry to climate,University of Michigan., 2013.
- Lindberg, J. D., Douglass, R. E. and Garvey, D. M.: Carbon and the optical properties of atmospheric dust, Appl.
  Opt., 32(30), 6077, doi:10.1364/AO.32.006077, 1993.
- 568 Madl, P., Yip, M., Ristovski, Z., Morawska, L. and Hofmann, W.: Redesign of a Thermodenuder and Assessment of
- its Performance Division of Molecular Biology Department of Physics and Biophysics Dosimetry and Modelling
   Working Group, (2001), 3065, 2003.
- 571 Majumder, A. K., Nazmul Islam, K. M., Bajracharya, R. M. and Carter, W. S.: Assessment of occupational and
- ambient air quality of traffic police personnel of the Kathmandu valley, Nepal; in view of atmospheric particulate
- 573 matter concentrations (PM10), Atmos. Pollut. Res., 3(1), 132–142, doi:10.5094/APR.2012.013, 2012.
- 574 Mauderly, J. L. and Chow, J. C.: Health effects of organic aerosols., Inhal. Toxicol., 20(3), 257–88,

- 575 doi:10.1080/08958370701866008, 2008.
- 576 Murugavel, P. and Chate, D. M.: Volatile properties of atmospheric aerosols during nucleation events at Pune, India,
- 577 J. Earth Syst. Sci., 120(3), 347–357, doi:10.1007/s12040-011-0072-7, 2011.
- Panday, A. K. and Prinn, R. G.: Diurnal cycle of air pollution in the kathmandu valley, nepal: Observations, J.
  Geophys. Res. Atmos., 114(9), 1–19, doi:10.1029/2008JD009777, 2009.
- Panday, A. K., Prinn, R. G. and Schär, C.: Diurnal cycle of air pollution in the Kathmandu Valley, Nepal: 2.
- 581 Modeling results, J. Geophys. Res. Atmos., 114(21), doi:10.1029/2008JD009808, 2009.
- 582 Pöschl, U.: Atmospheric aerosols: Composition, transformation, climate and health effects, Angew. Chemie Int.
  583 Ed., 44(46), 7520–7540, doi:10.1002/anie.200501122, 2005.
- 584 Putero, D., Cristofanelli, P., Marinoni, A., Adhikary, B., Duchi, R., Shrestha, S. D., Verza, G. P., Landi, T. C.,
- 585 Calzolari, F., Busetto, M., Agrillo, G., Biancofiore, F., Di Carlo, P., Panday, A. K., Rupakheti, M. and Bonasoni, P.:
- Seasonal variation of ozone and black carbon observed at Paknajol, an urban site in the Kathmandu Valley, Nepal,
   Atmos. Chem. Phys., 15(24), 13957–13971, doi:10.5194/acp-15-13957-2015, 2015.
- Ramanathan, V., Crutzen, P. J., Kiehl, J. T. and Rosenfeld, D.: Aerosols, climate, and the hydrological cycle.,
  Science, 294(5549), 2119–24, doi:10.1126/science.1064034, 2001.
- Regmi, R. P. and Maharjan, S.: Trapped mountain wave excitations over the Kathmandu valley, Nepal, Asia-Pacific
  J. Atmos. Sci., 51(4), 303–309, doi:10.1007/s13143-015-0078-1, 2015.
- 592 Robinson, A. L., Donahue, N. M., Shrivastava, M. K., Weitkamp, E. a, Sage, A. M., Grieshop, A. P., Lane, T. E.,
- 593 Pierce, J. R. and Pandis, S. N.: Rethinking organic aerosols: semivolatile emissions and photochemical aging.,
  594 Science, 315(5816), 1259–62, doi:10.1126/science.1133061, 2007.
- Ronai, Z. A., Gradia, S., El-Bayoumy, K., Amin, S. and Hecht, S. S.: Contrasting incidence of ras mutations in rat mammary and mouse skin tumors induced by anti -benzo[ c ]phenanthrene-3,4-diol-1,2-epoxide, Carcinogenesis, 15(10), 2113–2116, doi:10.1093/carcin/15.10.2113, 1994.
- 598 Sarkar, C., Sinha, V., Sinha, B., Panday, A. K., Rupakheti, M. and Lawrence, M. G.: Source apportionment of
- 599 NMVOCs in the Kathmandu Valley during the SusKat-ABC international field campaign using positive matrix
- 600 factorization, Atmos. Chem. Phys, 17, 8129–8156, doi:10.5194/acp-17-8129-2017, 2017.
- Schnaiter, M.: Absorption amplification of black carbon internally mixed with secondary organic aerosol, J.
  Geophys. Res., 110(D19), D19204, doi:10.1029/2005JD006046, 2005.
- Seinfeld, J. H. and Pandis, S. N.: Atmospheric Chemistry and Physics, Second., A Wiley-Interscience publication.,
   2006.
- 605 Shakya, K. M., Ziemba, L. D. and Griffin, R. J.: Characteristics and sources of carbonaceous, ionic, and isotopic
- species of wintertime atmospheric aerosols in Kathmandu valley, Nepal, Aerosol Air Qual. Res., 10(3), 219–230,
  doi:10.4209/aaqr.2009.10.0068, 2010.
- 608 Shakya, K. M., Rupakheti, M., Shahi, A., Maskey, R., Pradhan, B., Panday, A., Puppala, S. P., Lawrence, M. and
- Peltier, R. E.: Near-road sampling of PM2:5, BC, and fine-particle chemical components in Kathmandu Valley,
  Nepal, Atmos. Chem. Phys., 17(10), 6503–6516, doi:10.5194/acp-17-6503-2017, 2017.
- Sharma, R. K., Bhattarai, B. K., Sapkota, B. K., Gewali, M. B. and Kjeldstad, B.: Black carbon aerosols variation in
  Kathmandu valley, Nepal, Atmos. Environ., 63, 282–288, doi:10.1016/j.atmosenv.2012.09.023, 2012.
- Shiraiwa, M., Kondo, Y., Iwamoto, T. and Kita, K.: Amplification of Light Absorption of Black Carbon by Organic
  Coating, Aerosol Sci. Technol., 44(1), 46–54, doi:10.1080/02786820903357686, 2010.
- Shrestha, R., Kim, S.-W., Yoon, S.-C. and Kim, J.-H.: Attribution of aerosol light absorption to black carbon and
  volatile aerosols., Environ. Monit. Assess., 186(8), 4743–51, doi:10.1007/s10661-014-3734-5, 2014.
- Singh, A., Rajput, P., Sharma, D., Sarin, M. M. and Singh, D.: Black carbon and elemental carbon from postharvest agricultural-waste burning emissions in the Indo-Gangetic plain, Adv. Meteorol., 2014, doi:10.1155/2014/179301, 2014.
- 620 Slowik, J. G., Cross, E. S., Han, J.-H., Davidovits, P., Onasch, T. B., Jayne, J. T., Williams, L. R., Canagaratna, M.
- 621 R., Worsnop, D. R., Chakrabarty, R. K., Moosmüller, H., Arnott, W. P., Schwarz, J. P., Gao, R.-S., Fahey, D. W.,
- 622 Kok, G. L. and Petzold, A.: An Inter-Comparison of Instruments Measuring Black Carbon Content of Soot Particles,

- 623 Aerosol Sci. Technol., 41(3), 295–314, doi:10.1080/02786820701197078, 2007.
- 624 Sokolik, I. N. and Toon, O. B.: Incorporation of mineralogical composition into models of the radiative properties of
- 625 mineral aerosol from UV to IR wavelengths, J. Geophys. Res. Atmos., 104(D8), 9423–9444,
- 626 doi:10.1029/1998JD200048, 1999.
- 627 Stevanovic, S., Miljevic, B., Madl, P., Clifford, S. and Ristovski, Z.: Characterisation of a commercially available
- thermodenuder and diffusion drier for ultrafine particles losses, Aerosol Air Qual. Res., 15(1), 357–363,
- 629 doi:10.4209/aaqr.2013.12.0355, 2015.
- Warneck, P.: Chemistry of the natural atmosphere, Second edi., edited by R. Dmowska, R. J. Holton, and H. T.
  Rossby, Academic press., 2000.
- 632 Yu, H., Kaufman, Y. J., Chin, M., Feingold, G., Remer, L. A., Anderson, T. L., Balkanski, Y. and Bellouin, N.: A
- review of measurement-based assessments of the aerosol direct radiative effect and forcing, Atmos. Chem. Phys.,613–666, 2006.
- Zellner, R.: Global Aspects of Atmospheric Chemistry, edited by R. Zellner, Springer Science & Business Media.,
   1999.
- 637 Zhang, R., Khalizov, A. F., Pagels, J., Zhang, D., Xue, H. and McMurry, P. H.: Variability in morphology,
- 638 hygroscopicity, and optical properties of soot aerosols during atmospheric processing., Proc. Natl. Acad. Sci. U. S.
- 639 A., 105(30), 10291–6, doi:10.1073/pnas.0804860105, 2008.
- 640
- 641
- 642
- 643

644 Table 1. Summary of four sets of experiments carried out with their respective sampling dates

| S.N. | Experimental setup                                                        | Experiment date   |  |  |  |  |  |  |
|------|---------------------------------------------------------------------------|-------------------|--|--|--|--|--|--|
| 1    | Semi-volatile aerosol contribution to particle number                     | March-April, 2015 |  |  |  |  |  |  |
|      | concentration using CPC and thermodenuder setup                           |                   |  |  |  |  |  |  |
| 2    | Semi-volatile aerosol contribution to aerosol size                        | June, 2015        |  |  |  |  |  |  |
|      | distribution using SMPS and thermodenuder setup                           |                   |  |  |  |  |  |  |
| 3    | Semi-volatile aerosol contribution to total aerosol                       | April, 2015       |  |  |  |  |  |  |
|      | absorption using aethalometer and thermodenuder setup                     |                   |  |  |  |  |  |  |
| 4    | Semi-volatile aerosol contribution to total aerosol scattering July, 2015 |                   |  |  |  |  |  |  |
|      | using nephelometer and thermodenuder setup                                |                   |  |  |  |  |  |  |

| TDD set temp. in °C | Semi-volatile fraction of aerosol measured by CPC | Semi-volatile fraction of aerosol measured by |
|---------------------|---------------------------------------------------|-----------------------------------------------|
|                     | (%)#                                              | SMPS (%) #                         |
| Room temp.          | 12                                                | 20                                            |
| 50                  | 16                                                | 26                                            |
| 100                 | 18                                                | 32                                            |
| 150                 | 23                                                | -                                             |
| 200                 | 28                                                | 52                                            |
| 250                 | 46                                                | -                                             |
| 300                 | 49                                                | 62                                            |
|                     |                                                   |                                               |

**646 Table 2. Summary of semi-volatile aerosol fraction's physical properties at various temperatures.**

647 #The fraction represented in the table are derived from linear interpolation of slopes

| TDD set temp. | Loss of absorption at | Loss of absorption at | Absorption       | Average absorption   | Average absorption   | Average absorption   |
|---------------|-----------------------|-----------------------|------------------|----------------------|----------------------|----------------------|
| in °C         | 370nm (%) #           | 880 nm (%) #          | due to intrinsic | angstrom coefficient | angstrom coefficient | angstrom coefficient |
|               |                       |                       | properties,      | of wet aerosol *     | of dry aerosol       | of semi-volatile     |
|               |                       |                       | variation in     |                      | (Avg±SD)             | aerosol fraction     |
|               |                       |                       | size and brown   | (Avg±SD)             |                      | (Avg±SD)             |
|               |                       |                       | carbon           |                      |                      |                      |
| Room temp.    | 16                    | 16                    | 0                | 1.02±0.24            | 1.01±0.24            | 1.12±0.47            |
|               |                       |                       |                  |                      |                      |                      |
| 50            | 23                    | 20                    | 3                | $1.08 \pm 0.17$      | $1.08\pm0.18$        | 1.10±0.43            |
|               |                       |                       |                  |                      |                      |                      |
| 100           | 19                    | 18                    | 1                | 1.01±0.23            | 0.98±0.23            | 1.12±0.38            |
|               |                       |                       |                  |                      |                      |                      |
| 150           | 25                    | 21                    | 4                | $0.97 \pm 0.27$      | 0.92±0.30            | 1.19±0.41            |
|               |                       |                       |                  |                      |                      |                      |
| 200           | 31                    | 27                    | 4                | 0.97±0.19            | $0.92 \pm 0.18$      | 1.13±0.43            |
|               |                       |                       |                  |                      |                      |                      |
| 250           | 35                    | 28                    | 7                | 1.03±0.20            | 0.99±0.22            | 1.12±0.30            |
|               |                       |                       |                  |                      |                      |                      |
| 300           | 37                    | 28                    | 9                | 1.30±0.30            | 1.24±0.30            | 1.43±0.33            |

**648 Table 3.** Summary of influence of volatility on absorption at various temperatures.

649 \*Average absorption Angstrom coefficient of wet aerosols (ambient aerosol) while the simultaneous dry experiment was being

650 conducted at TDD set temperatures.

**The fraction represented in the table are derived from linear interpolation of slopes**

652

| 654 | Table 4. Summary of influence of | f volatility on scatteri | ng at various temperatures. |
|-----|----------------------------------|--------------------------|-----------------------------|
|     | 2                                | 2                        |                             |

| TDD set temp. | Loss of scattering | Loss of scattering | Loss of scattering | Average scattering   | Average scattering   | Average scattering   |
|---------------|--------------------|--------------------|--------------------|----------------------|----------------------|----------------------|
| in °C         | at 450nm (%) #     | at 550nm (%) #     | at 700nm (%) #     | Angstrom coefficient | Angstrom coefficient | Angstrom coefficient |
|               |                    |                    |                    | of wet aerosol *     | of dry aerosol       | of semi-volatile     |
|               |                    |                    |                    | (Avg±SD)             | (Avg±SD)             | aerosol fraction     |
|               |                    |                    |                    |                      |                      | (Avg±SD)             |
|               |                    |                    |                    |                      |                      |                      |
| Room temp.    | 18                 | 15                 | 8                  | 1.94±0.45            | 1.68±0.45            | 4.35±2.46            |
| 50            | 17                 | 13                 | 8                  | 1.76±0.38            | 1.47±0.34            | 5.52±2.01            |
| 100           | 29                 | 27                 | 20                 | $1.92\pm0.42$        | 1.69±0.39            | 2.85±0.32            |
| 150           | 39                 | 38                 | 32                 | 1.96±0.44            | $1.70\pm0.44$        | 2.69±0.71            |
| 200           | 48                 | 46                 | 40                 | 1.93±0.42            | $1.59 \pm 0.40$      | 2.65±0.64            |
| 250           | 62                 | 59                 | 52                 | 1.94±0.44            | $1.45 \pm 0.41$      | 2.61±0.46            |
| 300           | 71                 | 70                 | 66                 | 1.99±0.46            | $1.49\pm0.41$        | 2.47±0.80            |

655 \*Average absorption Angstrom coefficient of wet aerosols (ambient aerosol) while the simultaneous dry experiment was being

656 conducted at TDD set temperatures.

657 #The fraction represented in the table are derived from linear interpolation of slopes

**Table 5**. Summary of semi-volatile aerosol fraction Single Scattering Albedo (SSA) assuming wet aerosol SSA as 0.9 and 0.95 at different wavelengths and

TDD set temperatures.

| TDD set temp. in $^{\circ}C$ | Wavelength = 450nm   |                      | Wavelength = 550nm   |                      | Wavelength = 700nm   |                      |
|------------------------------|----------------------|----------------------|----------------------|----------------------|----------------------|----------------------|
|                              | Semi-volatile        | Semi-volatile        | Semi-volatile        | Semi-volatile        | Semi-volatile        | Semi-volatile        |
|                              | aerosol fraction     |
|                              | Single Scattering    |
|                              | Albedo (SSA) at      |
|                              | wet aerosol fraction |
|                              | SSA 0.9              | SSA 0.95             | SSA 0.9              | SSA 0.95             | SSA 0.9              | SSA 0.95             |
| Room temp.                   | 0.91                 | 0.95                 | 0.90                 | 0.95                 | 0.85                 | 0.92                 |
| 50                           | 0.86                 | 0.93                 | 0.83                 | 0.91                 | 0.78                 | 0.88                 |
| 100                          | 0.93                 | 0.97                 | 0.93                 | 0.97                 | 0.92                 | 0.96                 |
| 150                          | 0.94                 | 0.97                 | 0.94                 | 0.97                 | 0.94                 | 0.97                 |
| 200                          | 0.94                 | 0.97                 | 0.94                 | 0.97                 | 0.94                 | 0.97                 |
| 250                          | 0.95                 | 0.96                 | 0.95                 | 0.97                 | 0.95                 | 0.97                 |
| 300                          | 0.95                 | 0.98                 | 0.95                 | 0.98                 | 0.96                 | 0.98                 |

---

## Author Comment (AC5) · 27 Aug 2017

*Supplement of*

**Influence of semi-volatile aerosols on physical and optical properties of aerosols in the Kathmandu Valley**

**Sujan Shrestha[1,2], Siva Praveen Puppala[1], Bhupesh Adhikary[1], Kundan Lal Shrestha[2], Arnico K. Panday[1]**

*Correspondence to:* Siva Praveen Puppala (SivaPraveen.Puppala@icimod.org)

**S1: Calculation of Angstrom exponent of semi-volatile aerosol absorption/scattering**

As mentioned in the original manuscript, 'wet' sample always represents ambient aerosol and 'dry' sample represents ambient air passing through TDD. For better clarification, how we computed AAE and SAE, we provide below text as additional supplementary material.

The semi-volatile aerosol fraction contribution to ambient aerosol properties were measured through the difference between wet and dry aerosol properties.

$$SV_{AP} = WA_{AP} - DA_{AP}$$

Where,

$SV_{AP}$ = Semi-Volatile aerosol fraction contribution which can be number, scattering or absorption

$WA_{AP}$ = Wet aerosol property which is ambient aerosol number, scattering or absorption

$DA_{AP}$ = Dry aerosol property which is TDD derived aerosol number, scattering or absorption at different TDD set temperatures

For example semi-volatile aerosol fraction absorption contribution was calculated from the below formula.

$$SV_{Abs\_\lambda} = WA_{Abs\_\lambda} - DA_{Abs\_\lambda}$$

Where,

$SV_{Abs\_\lambda}$ = Semi-Volatile aerosol fraction absorption at wavelength $\lambda$

$WA_{Abs\_\lambda}$ = Wet aerosol absorption at wavelength $\lambda$

$DA_{Abs\_\lambda}$ = Dry aerosol absorption at wavelength $\lambda$

Wet and dry aerosol absorption were measured using identical aethalometers (AE-33) at seven different wavelengths. We derived semi-volatile aerosol fraction absorption at seven different wavelengths from above equation and aethalometer's (wet and dry) absorption data.

$$AE = -\frac{log\frac{E_{\lambda 1}}{E_{\lambda 2}}}{log\frac{\lambda_1}{\lambda_2}}$$

AE= Angstrom Exponent

$E_{\lambda 1}$ = Absorption/Scattering/Extinction coefficient at wavelength $\lambda 1$

$E_{\lambda 2}$ = Absorption/Scattering/Extinction coefficient at wavelength $\lambda 2$

From the above equation we derived wet, dry and semi-volatile aerosol fraction absorption/scattering angstrom exponent.

**S2. Calculation of Single Scattering Albedo (SSA)**

There is a constant fraction contribution of semi-volatile aerosol physical-optical properties in our experiments (figure 3, 7 and 10). Linear regression and correlation coefficients indicated that the average absorption and scattering losses at each temperature were almost consistent for a particular TDD set temperature with very little variation in the slope. Taking this into account, the linear slopes were used to derive the semi-volatile fraction contribution for wet (ambient) aerosol absorption and scattering. Same fractions were used to understand semi-volatile aerosol fraction contribution for given wet aerosols SSA. This will give important information on the nature of semi-volatile aerosol contribution to aerosol radiative forcing.

Single scattering albedo (SSA) is defined as the ratio of scattering to total extinction due to atmospheric aerosols as suggested in the equation below.

$$SSA = \frac{Scattering}{(Scattering + Absorption)} \tag{1}$$

Assuming wet aerosol SSA = 0.9 and scattering = 100, we derived the absorption using the above equation;

$$0.9 = \frac{100}{(Absorption + 100)} \tag{2}$$

$$=> Absorption = \frac{100 - 90}{0.9} \tag{3}$$

So, wet aerosol absorption = 11.11

Similarly, when we consider wet aerosol SSA = 0.95 and scattering = 100, absorption = 5.2

The semi-volatile aerosol fraction contribution derived from regression slopes were used in below equations.

$$Semi - volatile\ aerosol\ scattering = wet\ aerosol\ scattering *$$

$$(\%\ contribution\ of\ semi - volatile) \tag{4}$$

$Semi - volatile\ aerosol$ absorption $= wet\ aerosol$ absorption $*$

  $(\%\ contribution\ of\ semi - volatile)$       (5)

For wet aerosols scattering =100 and absorption=11.11

Semi-volatile aerosol scattering from equn. 4 = 24.58 (Table S1 Column 3, given below) (for

TDD set temperature 50°C while absorption = ((11.11*17)/100)) (Table S1 Column 2, given below)

$SSA = \dfrac{Semi-volatile\ aerosol\ scattering}{Semi-volatile\ aerosol\ scattering\ +\ Semi-volatile\ aerosol\ absorption}$   (6)

Semi-volatile SSA at 50°C = (24.58/(24.58+2.73))

  =0.861595 (Table S1 Column 4) (for TDD set temperature 50°C)

Where;

Scattering (%) = Loss of scattering at $T_i$

Absorption (%) = Loss of absorption at $T_i$

$T_i$ = TDD set temperature

Table S1: Table for wavelength interpolation

| At 450nm | TDD temp | Absorption fraction | Scattering Fraction | SSA of semi-volatile fraction assuming wet SSA=0.9 | SSA of semi-volatile fraction assuming wet SSA=0.95 |
|---|---|---|---|---|---|
| | Room Temp | 16.57 | 18 | 0.907291 | 0.954318 |
| | 50 | 24.58 | 17 | 0.861703 | 0.930072 |
| | 100 | 19.15 | 29 | 0.931707 | 0.966802 |
| | 150 | 23.05 | 39 | 0.938435 | 0.970183 |
| | 200 | 27.96 | 48 | 0.939269 | 0.970601 |
| | 250 | 30.36 | 62 | 0.948448 | 0.975169 |
| | 300 | 31.73 | 71 | 0.952738 | 0.977289 |
| At 550nm | | | | | |
| | Room Temp | 14.7 | 15 | 0.901892 | 0.951511 |
| | 50 | 23.59 | 13 | 0.832347 | 0.913776 |
| | 100 | 18.32 | 27 | 0.92996 | 0.965919 |
| | 150 | 22.02 | 38 | 0.939566 | 0.970749 |
| | 200 | 27.04 | 46 | 0.938748 | 0.97034 |
| | 250 | 29.13 | 59 | 0.948044 | 0.974969 |
| | 300 | 30.33 | 70 | 0.954112 | 0.977966 |
| At 700nm | | | | | |
| | Room Temp | 12.73 | 8 | 0.849886 | 0.923578 |
| | 50 | 20 | 8 | 0.782779 | 0.884956 |
| | 100 | 14.89 | 20 | 0.923668 | 0.962729 |
| | 150 | 18.33 | 32 | 0.940219 | 0.971075 |
| | 200 | 23.73 | 40 | 0.938218 | 0.970074 |
| | 250 | 25.64 | 52 | 0.948109 | 0.975001 |
| | 300 | 26.79 | 66 | 0.956887 | 0.979329 |

[Figure]

**Figure S1. (a)**  Comparison of collocated CPC particle concentration (CPC-1 and CPC-2 indicate the particle concentration (#/cm$^3$) measured in individual CPC instruments)

[Figure]

**Figure S1. (b)** Comparison of collocated Aethalometers black carbon concentration at 880 and 370nm (Aethalometer-1 and Aethalometer-2 indicate the black carbon concentration (μg/m$^3$) measured in individual

Aethalometers).

[Figure]

**Figure S2.** Leakage test conducted with CPC showing number concentration abruptly decreased to zero value in both instruments sampling wet and dry sample when HEPA filter is placed.

[Figure]

**Figure S3.** Diurnal variation of highly-volatile and moderately volatile aerosols.